



# On the cross-tropopause transport of water by tropical convective overshoots: a mesoscale modelling study constrained by in situ observations during TRO-Pico field campaign in Brazil

Abhinna K. Behera[1,2], Emmanuel D. Rivière[1], Sergey M. Khaykin[3], Virginie Marécal[4], Mélanie Ghysels[1], Jérémie Burgalat[1], and Gerhard Held[5]

[1]GSMA, UMR CNRS 7331, UFR Sciences Exactes et Naturelles, 51687 Reims CEDEX 2, France
[2]Now at Univ. Lille, CNRS, UMR 8518, Laboratoire d'Optique Atmosphérique (LOA), 59000, Lille, France
[3]LATMOS/IPSL, UVSQ Université Paris-Saclay, UPMC University Paris 06, CNRS, Guyancourt, France
[4]Centre National de Recherches Météorologiques, Université de Toulouse, Météo-France, CNRS, Toulouse, France
[5]Instituto de Pesquisas Meteorológicas (IPMet)/ Universidade Estadual Paulista (UNESP), Bauru, S.P., Brazil

**Correspondence:** Abhinna K. Behera (abhinna.behera@univ-lille.fr)

**Abstract.** Deep convection overshooting the lowermost stratosphere is well known for its role in the local stratospheric water vapour (WV) budget. While it is seldom the case, local enhancements of WV associated with stratospheric overshoots are often published. Nevertheless, one debatable topic prevails on the global impact of this event with respect to the temperature-driven dehydration of air parcels entering the stratosphere. As a first step, it is crucial to quantify their role at a local scale before
assessing their impact at a large-scale in a meteorological model. It would lead to a forcing scheme for overshoots in the global models.

   This paper reports on the local enhancements of WV linked to stratospheric overshoots, observed during the TRO-Pico campaign conducted in March 2012 in Bauru, Brazil, using the BRAMS (Brazilian version of RAMS) mesoscale model. Since numerical simulation depends on the choice of several preferred parameters, each having its uncertainties, we vary the
microphysics or the vertical resolution while simulating the overshoots. Thus, we produce a set of simulations illustrating the possible variations in representing the stratospheric overshoots. To resolve better the stratospheric hydration, we opt for simulations with the 800-m-horizontal-grid-point presentation. Next, we validate these simulations against the Bauru S-band radar echo tops and the TRO-Pico balloon-borne observations of WV and particles. Two of the three simulations' setups yield results compatible with the TRO-Pico observations. From these two simulations, we determine approximately $333\,\mathrm{t}$ to $2000\,\mathrm{t}$
of WV mass prevailing in the stratosphere due to an overshooting plume depending on the simulation setup. About 70% of the ice mass remains between the $380\,\mathrm{K}$ to $385\,\mathrm{K}$ isentropic levels. The overshooting top comprises pristine ice and snow, while aggregates only play a role just above the tropopause. Interestingly, the horizontal cross-section of the overshooting top is about $450\,\mathrm{km}^2$ at $380\,\mathrm{K}$ isentrope, which is similar to the horizontal-grid-point resolution of a simulation that cannot compute overshoots explicitly. These results could establish a forcing scheme of overshooting hydration or dehydration in a large-scale
simulation.



# 1 Introduction

Water vapour (WV), a component of the lower stratosphere, is exhibited extensively to be a part of the stratospheric chemistry, and consequently, in the ozone layer balance (Shindell, 1999, 2001; Herman et al., 2002). It is also an element of the polar stratospheric clouds (Toon et al., 1990). Apart from its chemical impact, WV is the primary greenhouse gas on the earth, mainly in the upper troposphere and lower stratosphere (UTLS). Furthermore, Solomon et al. (2010) report on the non-negligible variations in the surface temperatures due to minute changes in stratospheric WV at a decadal time scale.

The tropical belt establishes a gate where water enters the stratosphere (Brewer, 1949; Holton et al., 1995). At the first order, the supercooled temperature field across the tropical tropopause layer (TTL) drives the abundance of WV in the stratosphere (Holton and Gettelman, 2001; Randel et al., 2001). The TTL is a transition zone around the tropical tropopause extending from $14\,\mathrm{km}$ to $19\,\mathrm{km}$ with intermediate properties between the troposphere and the stratosphere (Folkins et al., 1999; Fueglistaler et al., 2009). Inside, beyond the level of positive radiative forcing, the air masses ascend slowly and get dehydrated by solid condensation or sedimentation of ice particles - known to be the cold-trap mechanism (Sherwood and Dessler, 2000). Several trajectory studies, overlooking the contribution of the deep convection in the TTL, can explain the abundance and variability of WV in the tropical tropopause, as observed via satellite-borne sensors confirming the cold-trap as the principal mechanism to dominate the entry of WV in the tropics (e.g., Fueglistaler et al., 2005; James et al., 2008). Nevertheless, the open-ended debates on the trend of the stratospheric WV during the 1990s and 2000s (Oltmans et al., 2000; Rosenlof et al., 2001; Randel et al., 2006; Scherer et al., 2008) and the tropopause temperature (Seidel and Randel, 2006) conclude that other factors can play a role in the processes of WV entering into the stratosphere (Randel and Jensen, 2013).

One identified factor is the deep convection in the tropics, overshooting the stratosphere. It injects ice particles directly above the tropopause, which may experience partial sublimation before falling back to the troposphere. Consequently, the net effect should be hydration that mitigates the large-scale dehydration effect. Recently many case studies of overshoots, both in modelling (e.g., Chaboureau et al., 2007; Grosvenor et al., 2007; Chemel et al., 2009; Liu et al., 2010; Dauhut et al., 2015) and based on the observations (e.g., Corti et al., 2008; Khaykin et al., 2009; Iwasaki et al., 2012; Sargent et al., 2014; Khaykin et al., 2016), validate the hydration effect of stratospheric overshoots at local scales in the tropical belt. Occasionally, studies also indicate that if the lower stratosphere is saturated to ice, the net effect results in dehydration by the ice crystals growth in the stratosphere, removing WV by the sedimentation (Hassim and Lane, 2010; Danielsen, 1982). The forward domain filling trajectory model (Schoeberl et al., 2018) establishes that the hydration process takes over the dehydration process at the tropopause level from December 2008 to February 2009. Schoeberl et al. (2018) also report on the 2% enhancement of global stratospheric WV in a numerical model simply by introducing deep convection. Nevertheless, the relative contribution of stratospheric overshoots over the cold-trap at a large or global scale remains uncertain.

In recent years, studies report that deep convection reaching the tropopause may influence the stratospheric WV budget on a large-scale. Subsequently, the deep convection is now a part of trajectory domain filling studies of stratospheric WV distribution





(e.g., Schoeberl and Dessler, 2011; Wright et al., 2011; Ueyama et al., 2015). Schoeberl et al. (2012) cannot rigorously conclude

on the quantitative characterisation of convective moistening of the stratosphere because of its small contribution. Furthermore, it is below the precision level of satellite $H_2O$ measurements. Nonetheless, Schoeberl et al. (2012) parameterise the impact of deep convection producing gravity waves to mitigate the TTL hydration. Ueyama et al. (2015) estimate an enhancement of $\sim 0.3$ ppmv of $H_2O$ across $100$ hPa at a large-scale in the southern hemisphere during the Austral summer of 2006-07 from a trajectory-based study; the trajectories are initialised from the satellite observed convective cloud tops. Advancing further,

Ueyama et al. (2018) report an enhancement of about $0.6$ ppmv WV at this level between 10°S-50°N during the 2007 Boreal summer. Carminati et al. (2014) obtain an indirect signature of the stratospheric overshoots at a global scale by studying the diurnal cycle of the EOS Aura MLS (Microwave Limb Sounder) $H_2O$ mixing ratio due to deep convection overshooting the $100$ hPa layer, highlighting the most active convective regions. However, by and large, no studies can quantify the critical impact of stratospheric overshoots on the global distribution of WV so far.

Another potential strategy to solve this problem is to upscale the stratospheric overshooting effects by forcing in a large-scale simulation, where the overshoots are perceived thoroughly in cloud-resolving numerical simulations. However, before considering this step, fine-scale numerical simulations of several cases must take place. The combined analysis of results would promote a forcing or parameterisation scheme of the stratospheric overshoot in a larger-scale simulation. The present paper aims at analysing such fine-scale simulations.

Here, we perform three simulations of an observational case of stratospheric overshoots using the BRAMS (Brazilian version of RAMS) mesoscale model. They are different from each other over the microphysical setup or the vertical grid structure. It produces a range of estimations on the ice injection into the stratosphere and the remaining after the sublimation. We use the data from a well-documented case on 13 March 2012 in Bauru, São Paulo State, Brazil, during the TRO-Pico, a small balloon campaign (Khaykin et al., 2016; Ghysels et al., 2016). On that day, two lightweight balloon-borne hygrometers intercepted a

hydrated stratospheric air parcel originating from two different overshooting plumes. However, the particle counter/backscatter sondes did not detect any ice particles then.

We organise the paper as follows: sect. 2 provides a brief overview of the observational case plus the TRO-Pico campaign and the balloon-borne instruments used for the WV measurements. Sect. 3 describes the BRAMS model and the setup of the three simulations. Sect. 4 is dedicated to the validations of the simulations using the TRO-Pico observational dataset. Sect. 5

discusses the main results depicting the structure and composition of the overshooting plumes. Sect. 6 comprises a quantitative investigation of the stratospheric WV mass budget. Finally, sect. 7 summarises the main results of this work and the strategies to an upscaling approach.

## 2   Observational case of 13 March 2012 at Bauru

### 2.1   Overview of TRO-Pico campaign

It was witnessed during the first intensive observation period (IOP1) of the TRO-Pico campaign in March 2012 in Bauru (22.36°S, 49.03°W), State of São Paulo, Brazil. The TRO-Pico is a French project funded by the Agence Nationale de la





Recherche (ANR), equipped with a small balloon campaign. It aims at investigating the impact of overshooting convection at a local and seasonal scale; also to learn deeper on its impact at a large-scale. During this campaign, several light-weight instruments were operated under $500\,\mathrm{m}^3$ and $1500\,\mathrm{m}^3$ open balloons, and under $1.2\,\mathrm{kg}$ rubber balloons. A large set of instru-

ments measuring $CO_2$, $CH_4$, $O_3$, and $NO_2$ also took part in this campaign. However, the principal instruments of the campaign were measuring WV and particles. On 13 March 2012, only the WV measuring instruments: Pico-SDLA and FLASH-B, and particles: LOAC and COBALD, were flown. Behera et al. (2018) detail the discussion of the second IOP of the TRO-Pico campaign during January-February 2013.

Pico-SDLA is an infrared laser hygrometer emitting at $2.61\,\mathrm{\mu m}$ in a $1\,\mathrm{m}$ long open optical cell (Ghysels et al., 2016). Its

uncertainty is about 4% in the TTL conditions. FLASH-B is a Lyman-alpha hygrometer measuring WV at night-time only with an uncertainty of 5% in the UTLS (Khaykin et al., 2009). LOAC is an optical particle counter based on the scattered light at $60°$ by ambient aerosol or particles for different wavelength channels (Renard et al., 2016). COBALD, developed at ETH-Zürich, is a backscatter sonde that applies several wavelengths (Brabec et al., 2012). Here, we use both the particles/aerosol instruments for the ice particle detection above the tropopause level.

## 2.2 Meteorological conditions, Flight trains, balloon-borne measurements

Before discussing the details of the observations, we summarise the meteorological conditions on March 13, 2012, in the central region of the State of São Paulo. This day was after the peak of the rainy season, with frequent heavy thunderstorms. There was no noticeable deep convective activity around Bauru before local noon (15:00 UT). The synoptic situation during the entire day exhibited an extremely weak pressure gradient across the complete São Paulo with very light westerly winds in the mid-levels

of the troposphere. Nonetheless, a vigorous thermodynamic instability prevailed throughout that afternoon. CAPE values of $\geq 4000\,\mathrm{J\,kg^{-1}}$ were forecasted by the $10\,\mathrm{km}$ horizontal resolution meso-ETA weather model in the central and western parts of São Paulo State. Meso-ETA is the model used for the met-briefing during the TRO-Pico campaign. These conditions were indeed favourable for the development of relatively small and short-lived deep convective cells, which started to appear from local noon. The main convective activity in the area of interest for the TRO-Pico campaign was about $100\,\mathrm{km}$ east of Bauru

near Botucatu, and later between Botucatu and Bauru with a series of short-lived and almost stationary convective cells. See sect. 4 and the animation on cloud tops in the supplementary material for the time evolution of the convective cells at these locations.

On 13 March 2012, a flight train comprising Pico-SDLA and LOAC sensors was launched at 20:20 UT under a $500\,\mathrm{m}^3$ Aerostar open balloon. The balloon reached the upper TTL around 21:54 UT and began to descend at around 22:00 UT under a

parachute from $\sim 24\,\mathrm{km}$ altitude. Three hours later, after the launching of Pico-SDLA, another flight train comprising FLASH-B and COBALD instruments was launched under a $1200\,\mathrm{g}$ Totex extensible balloon. This balloon burst at 23:39 UT. Ghysels et al. (2016) and Khaykin et al. (2016) report on the WV profiles from both stratospheric hygrometers. Within a layer from altitude $15\,\mathrm{km}$ to $21.2\,\mathrm{km}$, Ghysels et al. (2016) demonstrate a Pico-SDLA/FLASH Pearson correlation coefficient of 0.98, where both the hygrometers recorded two particular local enhancements of the WV mixing ratio at $18.5\,\mathrm{km}$ and $17.8\,\mathrm{km}$

altitude, respectively. Besides, they registered a third local enhancement at $17.2\,\mathrm{km}$ altitude, albeit of smaller magnitude in





comparison to the earlier two. One remarkable point is that the LOAC particle counter detected no ice particles within these altitudes during the flight train. Moreover, the COBALD backscatter sonde flown under the same balloon as FLASH ruled out the presence of ice particles.

The trajectory study of Khaykin et al. (2016) establishes a well-documented link between the local enhancement of WV in the stratospheric part of the TTL, seen by Pico-SDLA and FLASH-B, and the air mass advected from stratospheric overshooting plumes. However, the present work addresses further insights into the time evolution of this meteorological state based on a more extended analysis of a deep convective system developed during the local afternoon of 13 March 2012, in the southeast of Bauru and decayed in the evening. A comparison between Bauru S-Band radar images with model outputs is made in sect. 4 to monitor the detected convective activity and development of specific plumes.

**2.3   S-Band radar**

This modelling study benefits from the echo tops product of convective systems observed by the Doppler S-Band radar, located at IPMet/UNESP in Bauru. It facilitates the validation of our simulations. The echo top measurements depend highly on the technical specifications of the radar, such as wavelength, beam width, pulse width (PW), pulse repetition frequency (PRF), and radial and azimuth resolution. In the case of Bauru S-band radar, the beam width is $2°$; the PW is $0.8\,\mu s$ at PRF 620/465

pulses per second, limiting the range to $240\,km$ with a radial resolution of $250\,m$ and $1°$ in azimuth. Thus, the Bauru radar can only identify raindrops, liquid, or frozen particles, with a general threshold of $10\,dBZ$, corresponding to a rainfall rate of $0.15\,mm\,h^{-1}$ to $0.3\,mm\,h^{-1}$ when the beam cross-section is filled. The radar records reflectivity, spectral width, and radial velocities at 16 different elevations between $0.3°$ to $45°$. Due to the $2°$ beamwidth, it may underestimate the altitude and size of the overshooting plumes containing small cloud droplets and mostly ice particles when they are at a relatively long distance

from the station.

**3   BRAMS mesoscale model and simulation settings**

**3.1   Brazilian developments on the Regional Atmospheric Modeling System (BRAMS)**

BRAMS, version - 4.2, maintained at Centro de Previsão de Tempo e Estudos Climáticos (CPTEC) (Freitas et al., 2009), is a 3D regional and cloud-resolving model based on the RAMS model, version - 5.04, developed at Colorado State University

(CSU)/ATMET (Cotton et al., 2003). The Brazilian developments, tuned for the tropics, are essentially on the cumulus convection, surface scheme, and surface moisture initialisation. It simulates the turbulence, sub-grid scale convection, radiation, surface-air exchanges, and cloud microphysics with the 2-moment configuration at different scales ranging from large continental to large-eddy scale simulations. Furthermore, it can simulate seven types of hydrometeors, viz., cloud, and rain as liquid particles and pristine ice, snow, aggregate, hail, and graupel as ice particles (Walko et al., 1995). Here, the mixing ratios of hy-

drometeors and concentration are prognostic variables (Meyers et al., 1997). A gamma distribution represents all hydrometeors, where $\nu$, the shape parameter, determines both the modal diameter and the maximum concentration at that diameter.





$$f_{gam}(D) = \frac{1}{\Gamma(\nu)} \left( \frac{D}{D_n} \right)^{\nu-1} \frac{1}{D_n} exp \left( -\frac{D}{D_n} \right) \tag{1}$$

Equation. 1 is obtained from (Walko et al., 1995), where $f_{gam}$ denotes the probability density function for modified gamma distribution of hydrometeors with diameter $D$. $D_n$ is the characteristic diameter of the modified gamma distribution, and $\Gamma(\nu)$ is

the normalisation constant. A larger $\nu$ implies a larger modal diameter with a narrower distribution width. The BRAMS/RAMS can determine the fully compressible non-hydrostatic equations using a clever grid-nesting system that solves equations simultaneously between computational meshes applying any number of two-way interactions (Tripoli and Cotton, 1982). It also comprises a deep and shallow cumulus system based on the mass flux method (Grell and Dévényi, 2002), fit for simulating the convective transport of tracers. Using this model, Marécal et al. (2007) successfully reproduce the WV distribution in the trop-

ical UTLS in a deep convective atmosphere. Furthermore, Liu et al. (2010) simulate the stratospheric overshooting convection and corresponding WV enhancement in West Africa during the monsoon.

## 3.2   Simulation setups

We simulate three cloud-resolving simulations using the BRAMS model, implementing a multiple grid-nesting to resolve explicitly the stratospheric overshoots linked to the case study outlined in sect. 2. The strategy is to estimate the sensitivity

of the results to the model setup, producing various modelled hydration or ice injection schemes. It is likely to influence our inferences about the fundamental physical parameters linked to overshoots and the procedure to set them up in the forcing strategy of $H_2O$ into the lower stratosphere.

Subsequently, we perform simulations with a spatial resolution of $800\,\mathrm{m} \times 800\,\mathrm{m}$ in the third grid of the grid-nesting system. We consider the first simulation as the reference simulation, denoted as REF hereafter. The second simulation deviates from

REF with respect to the shape parameter of the hydrometeors in the bulk microphysics setting, indicated as NU21 ($\nu = 2.1$) hereafter. The third simulation has a higher vertical grid-point presentation to determine better the TTL dynamics than REF, referred to as HVR, High Vertical Resolution, hereafter. Then, we examine the sensitivity to the microphysical context in NU21 and the vertical resolution in HVR, both having an expected impact on the simulation of deep convection and overshooting plumes.

## 175   3.2.1   General setup

REF, NU21, and HVR comprise the grid-nesting system of three grids holding the same grid positions and the same horizontal grid-point presentation. The horizontal grid-point resolution increases from $20\,\mathrm{km}$, parent grid, to $4\,\mathrm{km}$ in the second grid and $800\,\mathrm{m}$ in the third grid. The parent grid encompasses a large part of southern Brazil with a domain of $1840\,\mathrm{km} \times 1640\,\mathrm{km}$, centred at 23°S, 49.9°W. The second grid comprises a domain of $964\,\mathrm{km} \times 624\,\mathrm{km}$, encompassing the state of São Paulo,

centred at 22.4°S, 49.0°W, slightly south of Bauru. The area of the third grid covers the most active convective region around Bauru with a domain size of $201\,\mathrm{km} \times 165\,\mathrm{km}$, centred at 22.1°S, 49.2°W. Note that irrespective of the vertical resolution of



the simulations, we restrict the top layer of the domain to $30\,\mathrm{km}$ altitude with a sponge layer of $5\,\mathrm{km}$ to absorb gravity waves at the top.

Each simulation starts at 12:00 UT on 12 March 2012 and finishes $48\,\mathrm{h}$ later. To make it cost-effective computing, we
activate the third grid only at 10:00 UT on 13 March and record afterwards the model outputs every $7.5\,\mathrm{min}$. This data record frequency equals the volume scans generated by IPMet S-band radar. These are used for validating the modelled cloud tops. The simulation integration time step varies to ensure numerical stability, which is $2\,\mathrm{s}$ to $10\,\mathrm{s}$ for the first grid. It is five times less for the second grid and 25 times less for the third grid. The time resolution of invoking the radiation module is $300\,\mathrm{s}$ to $500\,\mathrm{s}$. The ECMWF operational analyses initialise all the simulations, also force the boundary conditions of the first grid every
$6\,\mathrm{h}$. Note that there is no nudging of ECMWF data at the centre of the domain - following the work of Liu et al. (2010).

### 3.2.2 Specific setup

REF, NU21, and HVR simulations deviate from each other over the following points.

- The shape parameter ($\nu$) in the gamma function distribution concerning the hydrometeors is $\nu = 2.0$ in REF, however, it is $\nu = 2.1$ in NU21. On 13 March 2012, at 10:00 UT, we introduce this setting to all the grids of NU21. Both NU21
and REF are exactly equal until this point in time. The aim here is to study the impact of this microphysical parameter during the most active time of deep convection and avoid any possible early divergence.

- HVR differs from REF with respect to the vertical grid-point resolution in the TTL. REF has 68 vertical levels with about $300\,\mathrm{m}$ resolution within the TTL whereas, HVR has 99 vertical levels with typically $150\,\mathrm{m}$ vertical resolution within the TTL, except at the tropopause level where it is $100\,\mathrm{m}$. Unlike REF and NU21, HVR is carried out entirely at the higher
vertical resolution starting at 12:00 UT on 12 March 2012. In the BRAMS model, it is unfeasible to change the vertical grid structure in the middle of the integration of simulation unless each layer in REF would correspond to a layer in HVR, which is not the case here.

## 4 Validation of the simulations

We validate all the three BRAMS simulations using observations from the S-Band radar of IPMet, located in Bauru, and the
balloon-borne measurements of the TRO-Pico campaign, respectively. Note that the balloon-borne measurements are part of the IOP1 phase of the two-year field campaign.

### 4.1 Validation of modelled cloud tops against radar echo top observations

We assess the ability of the BRAMS model in triggering and describing the deep convection activity at the accurate time and position by interpreting the modelled outputs versus the S-Band radar measurements. To do so, we estimate the modelled
cloud top layers every $1\,\mathrm{km}$ varying from $9\,\mathrm{km}$ to $20\,\mathrm{km}$ altitude, in a very similar manner to the echo top products. In the simulations, if the condensed water, i.e., ice plus liquid, concentration is above a prescribed threshold of mixing ratio within



a given layer, we determine the cloud top for this range of altitude. As we implement this criterion in a bottom-top loop, the cloud top altitude assignment for a given (x, y) grid mesh is definitive once all the vertical levels are browsed. To account for the decrease in the concentration of hydrometeors with altitude inside the TTL linked to a deep convective cell, we implement

the threshold of condensed water concentration to a cloud top depending on its range of altitudes. It is $1\,\mathrm{g\,kg^{-1}}$ for the layer starting from $9\,\mathrm{km}$ to $10\,\mathrm{km}$ up to the layer of $15\,\mathrm{km}$ to $16\,\mathrm{km}$. Beyond this, it is $0.45\,\mathrm{g\,kg^{-1}}$ for $16\,\mathrm{km}$ to $17\,\mathrm{km}$, $0.2\,\mathrm{g\,kg^{-1}}$ for $17\,\mathrm{km}$ to $18\,\mathrm{km}$, and $0.008\,\mathrm{g\,kg^{-1}}$ for the layers above $18\,\mathrm{km}$. We select these thresholds as a function of typically modelled concentrations of hydrometeors within overshooting plumes (see Liu et al., 2010).

Fig. 1 allows a qualitative comparison of the radar echo tops and modelled cloud tops from the three simulations. It illustrates

the capacity of BRAMS to reproduce the principal features: triggering deep convection, structure evolution, and severity of the overshooting plume in this relatively unorganised convective cluster. Note that here we compare the convective plumes when they are within $100\,\mathrm{km}$ radius of Bauru (the inner circle in Fig. 1a) to avoid a relatively large scanning angle of the radar, and thus to obtain accurate echo top heights. Furthermore and importantly, the modelled cloud tops are well within the third grid, not near or at the edges of this grid. We observe that the model can reproduce relatively well these highly unpredictable

convective systems. There exist similar deep convective clusters around Bauru in the radar images and the simulations, although at slightly different times. The radar image at 16:46 UT (13:46 Local Time; Fig. 1a) shows a storm cluster comprising three cells near Botucatu, southeast of Bauru with the echo top of the furthest west one reaching $>18\,\mathrm{km}$. On the height of radar echo tops, we should note that radar is more sensitive to liquid droplets than ice particles, which is the main component of overshooting plumes. In REF (Fig. 1b), we notice a comparable convective storm complex to have developed at 16:15 UT west

of Bauru, depicting two tops of $>17\,\mathrm{km}$ and $>18\,\mathrm{km}$, respectively. NU21 (Fig. 1c) shows a similar convective system at 15:45 UT in the west of Bauru, as seen in the radar in the southeast of Bauru one hour earlier (Fig. 1a), but with only one cloud top $>18\,\mathrm{km}$. HVR (Fig. 1d) also generates a convective cluster at 15:45 UT in the west of Bauru, but comprising three cells in the proximity of Bauru, $100\,\mathrm{km}$, with two cloud tops of $>17\,\mathrm{km}$ and one of $>18\,\mathrm{km}$.

The full-time series of the comparison between the modelled cloud tops and the S-band radar echo tops is in the supplemen-

tary material (animation of cloud tops) every $7.5\,\mathrm{min}$ from 15:01 UT to 18:52 UT on 13 March 2012. Fig. 1 demonstrates the main features of this series of comparison at the peak of the convective activity. At the beginning of the convective activity (15:01 UT), the radar is nearly cloud-free; the only storm cells are now at about $100\,\mathrm{km}$ south-southeast of Bauru near Botucatu with typically $9\,\mathrm{km}$ to $10\,\mathrm{km}$ altitude. REF reproduces this feature qualitatively with the same range of maximum height but much closer to Bauru, however, at the south/northwest of Bauru. The same type of storm cluster is observed in NU21 at 14:15

UT. About $45\,\mathrm{min}$ later, at 15:00 UT, NU21 produces convective activity triggering at the same position as in REF but with more intensity and higher cloud tops. It highlights that deep convection triggers earlier in NU21. Now, at 14:15 UT, there is no sign of convective activities in HVR, unlike in the radar image, but it appears at 15:00 UT near Ourinhos - southwest of Bauru. The storm cells are now overgrown in the area than in NU21 at 14:15 UT, though in a similar position. By the time HVR reaches 15:00 UT, the deep convection altitude is also higher than in REF and the radar echo tops. It is also located

much more west than the radar observations. However, stratospheric overshoots are present in the simulations as well as in the radar observations with the echo top above $17\,\mathrm{km}$ at the peak of the convective activity, i.e., during 16:00 - 17:00 UT. The



(a) radar observation at 16:46 UT (13:46 LT)

(b) REF

(c) NU21

(d) HVR

**Figure 1.** Snapshots of echo tops, observed by the S-band radar and modelled cloud tops from the BRAMS simulations on March 13, 2012, centred at Bauru. (a) radar observation at 16:45 UT, (b-d) REF at 16:45 UT, NU21 at 15:45 UT, and HVR at 15:45 UT, respectively. The circle displayed in panels b, c, and d corresponds to the 240 km radar range in panel a.

convective activity becomes greater in height and spread over larger areas in the TTL with time in the three simulations. It is





more to the west-southwest of Bauru in HVR. Thus, all simulations show the onset of convective activity slightly earlier than the observations. Note that it is an entirely complex task to associate one-by-one model and radar overshooting plumes and

might not be an ideal criterion to assess these simulations of unorganised deep convective clouds.

Here, we tabulate (Table 1) the number of overshooting plumes higher than $17\,\mathrm{km}$ altitude - the radar criterion of detecting overshoots, during the period 15:00 - 18:30 UT on 13 March 2012 within $100\,\mathrm{km}$ radius of Bauru. The observation time is confined until 18:30 UT, as the radar images show the decay of deep convection afterwards.

Table 1 depicts that REF can produce an equivalent number of overshooting plumes detected by the S-Band radar, although

slightly higher in altitudes. We anticipate this since the radar sensitivity to low-ice content is weak, following which the radar may underestimate the number of overshoots. Besides, another reasonable explanation would be a scenario where the $380\,\mathrm{K}$ layer may lie beneath the threshold of $17\,\mathrm{km}$ altitude. Then, the total number of overshooting cells in NU21 indicates that it is less intense in producing overshooting plumes than REF and radar. The time series analysis of cloud clusters shows that the lifetime of the overshooting plumes seems to be longer than REF, where the overshooting plumes reach $19\,\mathrm{km}$ rarely (see the

animation on cloud tops in the supplementary material). In HVR, however, the number of overshooting plumes is considerably high than REF and NU21.

Conceptually, with the composition of high vertical resolution, we expect HVR to determine more reliable dynamics across the tropical tropopause while compared with REF and NU21, respectively. Yet, contrary to the expectations, it tends to intensify significantly deep convection activity. A plausible fact to explain such behaviour in HVR would be the ratio between the vertical

and the horizontal grid points, which overestimates the vertical motions because of the grid cell saturation (see Homeyer et al., 2014; Homeyer, 2015). Moreover, high vertical resolution Eulerian model simulations can determine the high-frequency wave motions, e.g., inertial gravity wave. Consequently, they can produce unrealistic cloud conditions around the TTL (Jensen and Pfister, 2004) by overestimating cloud microphysics (see Aligo et al., 2009). As a result, we omit HVR to discuss the details in the succeeding sections and do not analyse this simulation for the water budget in the lower stratosphere.

In sect. 4.1, we essentially outline several principal aspects by closely studying the simulated convective plumes. First, we locate the position of deep convective activity further west/northwest in the model, typically $50\,\mathrm{km}$ to $60\,\mathrm{km}$ west/northwest. Second, the time evolution of the convective clusters reveals that they are moving north/northwest while most of the convective activity remains in the west of the Tietê river in both cases. Overall, we cannot expect the model to predict precisely the position and time of convective activity development. REF and, to a certain extent, NU21 provide reasonable predictions in space and

time. They generate good estimates of convective cloud tops but initiates the plumes generally earlier to the radar observation. In contrast, HVR yields unfavourable conditions and exaggerates its size.

## 4.2 Validation against TRO-Pico balloon-borne measurements

The WV and particle measurements performed in the proximity of overshoots in the frame of the TRO-Pico campaign establish a well-documented database to validate model simulations. For our study, as the balloon-borne measurements belong to a

moment several hours after the overshooting event - this time interval between the overshooting event and the balloon-borne measurements is indicated as $\delta t_{om}$ hereafter, the simulation validation strategy is as follows. We observe the modelled over-



shooting plume at $17.2\,\mathrm{km}$ and $17.8\,\mathrm{km}$ altitudes, respectively, where FLASH-B and Pico-SDLA hygrometers captured the WV local enhancements (see Khaykin et al., 2016). Then, after the same $\delta t_{om}$, we investigate the WV enhancement at these levels in the model.

### 4.2.1 REF simulation

To validate the local WV enhancement at $17.2\,\mathrm{km}$ altitude due to the modelled overshoots, we combine the TRO-Pico measurements by FLASH-B at 23:45 UT corresponding to an overshooting event that occurred at 16:46 UT with $\delta t_{om} = 7\,\mathrm{h}$ on 13 March 2012. We observe the time evolution of the modelled (REF) overshooting plume at $17.2\,\mathrm{km}$ altitude from 16:15 UT until 23:15 UT to maintain the same $\delta t_{om}$. Fig. 2 illustrates the horizontal cross-section of the total water content at $17.2\,\mathrm{km}$

altitude at three different time steps, viz., 16:15 UT, 19:45 UTC, and 23:15 UT, sequentially. It also draws the horizontal wind streamline to follow the direction of the moving plume at this height. We prepare this kind of plot every $7.5\,\mathrm{min}$ to follow the evolution of the overshooting cell at that height. For simplicity and space limitations, we show only these three plots in the paper.

Fig. 2a illustrates REF determined overshooting plume at 22.2°S, 49.15°W entering the stratosphere at 16:15 UT. About

after $3.5\,\mathrm{h}$, we observe this plume spreading wide horizontally (Fig. 2b), mostly east to 49.4°W. Furthermore, several other overshooting plumes developed in between but did not interact with the eastern part of the convective plume. Around 23:15 UT (Fig. 2c), most of the original plume moved eastward of 49.1°W by advecting northward, as precisely as described in the trajectory analysis of the same case in Khaykin et al. (2016). At some positions within the overshooting plume corresponding to the maxima of $H_2O$ mixing ratio (ice + liquid + vapour), we obtain the local enhancement is typically $2\,\mathrm{ppmv}$ of the total

water content (see Fig. 3a) at this altitude within $\pm 35\,\mathrm{km}$ northeast of Bauru.

Fig. 3 highlights such $H_2O$ enhancement domains in isolines. In Fig. 3a, at 23:15 UT around Bauru within an area of $70\,\mathrm{km} \times 50\,\mathrm{km}$, tilting northeast following the analysis in Fig. 2, REF produces many grid-points representing $H_2O$ enhancement of about $0.5\,\mathrm{ppmv}$ at $17.2\,\mathrm{km}$ altitude, which is in agreement with FLASH-B and Pico-SDLA measurements. The confirmation of no ice remaining indicates that all the ice has sublimated or sedimented in the simulation. It agrees with the

measurements carried out using LOAC and COBALD under the Pico-SDLA and FLASH-B, where they did not detect any ice particles in the stratosphere. The modelled $0.5\,\mathrm{ppmv}$ enhancement at $17.2\,\mathrm{km}$ level is comparable to the one measured by FLASH-B, $0.45\,\mathrm{ppmv}$, in that range of altitude. REF also produces very high $H_2O$ enhancement, $>10\,\mathrm{ppmv}$, in the northwest region away from Bauru. Such extremely wet conditions are possible due to a very recent overshoot in this area in the simulation.

Then, we implement the same strategy to validate the hydration due to overshoot at $17.8\,\mathrm{km}$ altitude, see Fig. 3b. It is the altitude of the second water enhancement captured by both Pico-SDLA and FLASH-B hygrometers. Khaykin et al. (2016) report this $H_2O$ enhancement comes from another overshooting plume than the one explaining the $17.2\,\mathrm{km}$ $H_2O$ enhancement. We investigate if a realistic overshooting plume in BRAMS can appear with a similar $H_2O$ enhancement following the same $\delta t_{om}$ time around Bauru. For the $H_2O$ enhancement at $17.8\,\mathrm{km}$ altitude identified by Pico-SDLA at 22:04 UT, the associated over-

shooting event occurred at 17:38 UT. This implies the $\delta t_{om} = 4\,\mathrm{h}26\,\mathrm{min}$. Following the overshooting plume, as in Fig. 2, from



16:15 - 20:52 UT, REF yields a similar $\delta t_{om}$ while obtaining the $H_2O$ enhancement. REF produces many grid-points/pixels with $H_2O$ enhancement of $0.7\,$ppmv around Bauru within an area of $70\,$km $\times\,50\,$km. Some pixels show more than $2\,$ppmv of $H_2O$ enhancement. Here, it is notable that BRAMS compute no ice in this part of the plume, which is in agreement with the COBALD and LOAC measurements. The $0.7\,$ppmv local enhancement at $17.8\,$km is thus fully compatible with the one measured by Pico-SDLA, $0.65\,$ppmv, and by FLASH-B, $0.55\,$ppmv, at this altitude.

The purpose of the investigation is to witness the same order $H_2O$ enhancement in the model corresponding to the TRO-Pico campaign measurements. And the approach of selecting an area of $70\,$km $\times\,50\,$km tilting northeast direction around Bauru is to consider only the $H_2O$ enhancement within this area (see Fig. 3). Furthermore, it corresponds to the point that the overshooting cells at $17.2\,$km and $17.8\,$km heights, respectively, are induced by two separate overshooting plumes (Khaykin et al., 2016).

### 4.2.2 NU21 simulation

With the same validation approach, as in REF, we select the overshooting plume that occurred at 16:15 UT in NU21. We study the time evolution of the overshooting plume at $17.2\,$km altitude from 16:15 UT to $(16:15 + \delta t_{om})$ UT, that is 23:15 UT - $\delta t_{om}$ is $7\,$h from the overshooting event till the FLASH-B measurement. It is similar as in Fig. 2 and is provided in the supplementary material (Fig. S1). The plume spreads horizontally, slightly southeastward, and finally northward, where most of the original plume is north to $22.4°$S and east to $48.8°$W at 23:15 UT.

In Fig. 4a, the conclusions are similar to REF. The total water content at $17.2\,$km altitude at 23:15 UT shows an enhancement of several ppmv, up to $2\,$ppmv at certain positions, particularly at the core of the plume. Many pixels within $10\,$km neighbourhood of Bauru show the WV enhancement of half a ppmv near the border of the overshooting plume. It is compatible with the local enhancement measured by FLASH-B at $17.2\,$km height. Moreover, it is crucial to recall the evidence of no ice remaining at this level, and the total $H_2O$ is only in the vapour phase as observed by the LOAC particle counter and the COBALD backscatter sonde. Then, we analyse the WV enhancement in NU21 at $17.8\,$km altitude at 20:52 UT, that is $\delta t_{om} = 4\,$h$40\,$min after the 16:15 UT overshooting event (see Fig. 4b). This $\delta t_{om}$ is the same as the time interval between the Pico-SDLA measurement and the overshooting event. In Fig. 4b, we obtain many pixels, located at the border of the overshooting plume, with a $\sim 0.7\,$ppmv $H_2O$ enhancement without any ice remaining - a very similar way of observation by Pico-SDLA and LOAC. Furthermore, there are many pixels near the Tietê River giving very high WV enhancement, up to $6\,$ppmv. This sort of large water enhancement from overshoots has already been identified by the FISH hygrometer onboard the Russian M55 Geophysica high-altitude aircraft in the SCOUT-AMMA field campaign in West Africa (see Schiller et al., 2009). It is now reasonable to state that BRAMS simulated overshooting plumes responsible for the local WV enhancements; however, not necessarily exactly in the same locations as the observed ones. Moreover, the wind spreading about the overshooting plumes are somewhat different from the realised ones during the TRO-Pico field campaign.

### 4.3 Conclusion of the Validation

In sect. 4.2, we demonstrate that the BRAMS model, via REF and NU21, can simulate fairly realistic deep convective plumes that are compatible with the IPMet S-Band radar observation during the temporal evolution of deep convective cloud systems





over three hours. However, these modelled deep convective plumes slightly west/northwest of the radar observation but with an
intensity comparable to the detected ones by the S-band radar. Furthermore, the convective cloud tops are sometimes higher in
altitude than the radar images. It corresponds to a possible fact that the S-Band radar is a little sensitive to the ice hydrometeors
- the main component of overshooting plumes addressed in subsequent sections. The number of overshooting plumes above
$17\,\mathrm{km}$ is comparable both in the model and the S-band radar images until 18:30 UT, after which the model exhibits convec-
tive activity with a longer lifetime. The study of overshooting plumes at $17.2\,\mathrm{km}$ and $17.8\,\mathrm{km}$ altitude, respectively, and the
corresponding total water enhancements after $\sim4.5\,\mathrm{h}$ and $7\,\mathrm{h}$, respectively, agree with both the balloon-borne measurements
of $H_2O$ mixing ratio by Pico-SDLA and FLASH-B hygrometers. Moreover, note that the grid-points showing several ppmv of
total $H_2O$ enhancement are often at the edge of the overshooting turret - coherent with the trajectory analysis of Khaykin et al.
(2016), reporting that the air masses sampled by the balloons are at the edge of the plume coming from the overshoot.

Thus, this study puts to the fore that fine-scale simulations using the BRAMS model can reproduce the overshooting con-
vection. Both REF and NU21 can lead now to more insight into the overshooting plumes within unorganised deep convective
plumes. Certain standard features like the amount of ice injection, width and surface area of the plume, $H_2O$ mass flux, and the
lifetime of the active cell, which we cannot measure directly with the current possible resources. In the subsequent sections,
we give a quantitative interpretation of the overshooting plumes from REF and NU21. Unfortunately, HVR appears to produce
excessively severe convective activity, making it unsuitable for further analysis.

## 5   Analyses of overshooting turret

We document the quantitative information retrieved on the structural aspects of a conventional overshooting plume from the
simulations. It can lead to a potential prescription of an explicit forcing scheme of overshoots in a mesoscale model. As such,
it may lead to the quantification of the influence of overshoots on a large-scale.

We present the five possible combinations of hydrometeors inside an overshooting plume. We choose its base to be at $380\,\mathrm{K}$
isentropic level, i.e., the lowest layer of the stratosphere. Furthermore, the instantaneous mass flux of individual hydrometeors
at $380\,\mathrm{K}$ isentropic level is estimated. It includes the estimation of total ice mass and the five types of ice particles between
$380\,\mathrm{K}$ to $430\,\mathrm{K}$ isentropic levels. In the end, a table gives the quantities that may lead to a blueprint - a forcing scheme of the
water vapour enhancement in the lower stratosphere due to overshoots in the numerical models.

### 5.1   Structure and composition of overshoots

We assess all the five types of ice hydrometeors during an overshooting event. The series of plots in Fig. 5 represents the
horizontal cross-section of the ratio of different ice hydrometeors over the net ice varying with altitude around the TTL in the
vicinity of the overshooting event that occurred at 16:15 UT in REF (see Table 1). We present this calculation from 15:00 -
18:52 UT for REF and NU21, which can be found in the animation of horizontal cross-sections in the supplementary materials.

We find pristine ice and snow to be the principal ice hydrometeors above the tropopause level ($\sim16.6\,\mathrm{km}$ altitude). However,
aggregates and a small amount of graupel are present to some extent. It is the case only for REF; see the horizontal cross-section





animation in the supplementary material for the full-time evolution. The presence of hail particles is negligible, not shown in Fig. 5, which confirms the results of Homeyer and Kumjian (2015), obtained using the S-band radar measurements of deep convective activities over extratropics. Furthermore, it is compatible with the results reported in Chemel et al. (2009); they study the Hector thunderstorm using the WRF model and obtain (pristine) ice and snow as the prime components. However,

the current work takes advantage of the BRAMS model combining five types of ice hydrometeors instead of three in the WRF version used by Chemel et al. (2009). Fig. 5 also depicts the non-negligible proportion of aggregates and graupel at the tropopause level within the overshooting plume, particularly for REF. Remarkably, pristine ice is absent entirely at the base of the turret (16.6 km altitude, $\sim$380 K) near the deepest core of the plume. Only snow, aggregates, and to some extent, graupels prevail. In NU21, it is mostly snow particles remaining as dominant ice hydrometeors spreading in a narrow area of a radius

of about 5 km. Moving further upward from the tropopause level, we observe just pristine ice (70%) and snow (30%) as the principal constituents of the overshooting dome reaching up to 18 km, well in the stratospheric part of the TTL. REF includes a small amount of graupel and aggregates, which is not the case for NU21. Interestingly, the constituent of the overshooting dome is 100% the pristine ice located at the edge of the plume around the tropopause level in all three simulations.

Next, we determine the contact area or the spreading (km$^2$) of the overshooting plume at the lowest layer of the stratosphere,

i.e., 380 K isentropic level. Fig. 6 illustrates the spreading of overshooting plumes at 380 K isentropic level for REF and NU21 at different time steps followed from Table 1.

A significant result from Fig. 6 is the average surface area of the spreading of the overshooting plume at 380 K level is about 450 km$^2$. It is approximately the grid-box area of a large-scale simulation (400 km$^2$), and Behera et al. (2018) shows that with such horizontal grid resolution, BRAMS cannot represent overshoots explicitly. They communicate about the TTL dynamics

and WV variability at a continental scale during a complete wet season by not allowing the overshoots in a simulation using a horizontal resolution of 20 km $\times$ 20 km, i.e., 400 km$^2$.

Furthermore, we compare the horizontal spreading between REF and NU21. In Fig. 6, the upper panel represents the surface areas of RFE, which are of 11 km $\times$ 15 km at 1537 UT and 22 km $\times$ 24 km at 16:37 UT, respectively. In the case of NU21, the lower panel, the surface areas are of 22 km $\times$ 24 km and 11 km $\times$ 11 km at 15:30 UT, and 30 km $\times$ 41 km at 15:52 UT,

respectively. The latter one with the large surface area indicates that changes in the particle size distribution, the shape parameter $\nu$, may modulate the spreading of overshooting convection while penetrating the stratosphere. In the following sections, we estimate the mass budget corresponding to UTLS, set as a preferred range of isentropic levels.

## 6 Stratospheric Water mass budget

We estimate each hydrometeor's instantaneous mass-flux rate across the 380 K isentropic level. The rates are the average

over the domain that comprises only the third grid of simulation. Please note that it is not representative of a property of any particular overshooting plume but preferably addresses a realistic estimation on the flux rates of ice particles entering the 380 K isentropic layer. Besides, we evaluate the net H$_2$O mass budget prevailing within the slice of 380 K to 430 K isentropic levels.



## 6.1 Mass-flux across 380 K isentropic level

Fig. 7 presents the domain-average instantaneous mass-flux rate for REF and NU21 across the 380 K isentropic level over the
third grid of the simulation during 14:00-18:52 UT. It depicts primarily the inferences drawn from Fig. 5. Such as the principal
hydrometeors are pristine ice and aggregates and to a much lower amount of snow and graupel, where the order of magnitude of
the maximum mass-flux rate of snow and graupel is about four-fold smaller than the maximum of pristine ice and aggregates.
Albeit the non-negligible mass-flux rate of graupel, its ratio in the structure of the overshooting turret remains modest. It occurs
approximately 10% of the composition of overshooting plume in a limited area only in REF exceeding the tropopause level
($\sim16.6$ km). Then, we associate the contrast in the snow composition inside the plume with the sedimentation. Graupel, denser
than snow, falls faster to the troposphere, results in the accumulation of snow in the stratosphere. Though the overshoots begin
at different times in REF and NU21, the local maximum of mass-flux rates are of the same order of magnitude, and in REF,
it is regularly higher than NU21. It is already explicit that the number of overshooting events is different in REF and NU21
(please refer to Table 1). Eventually, the differences in the mass-flux rate between REF and NU21 would be critical to explain
as their values are also proportional to the vertical wind velocity (see Sang et al., 2018).

## 6.2 Mass budget above 380 K isentropic level

Fig. 8 illustrates the total mass budget corresponding to the five types of ice hydrometeors, including water vapour. The
estimation is limited to the third grid only of the simulations between the 380 K to 430 K isentropic levels. The upper level is
430 K because none of the convective plumes overshoots this level in the simulations.

The total $H_2O$ (ice + liquid + vapour) mass budget estimations with respect to the unperturbed state - the time before the
start of deep convection in each simulation which is 15:00 UT for REF and 14:00 UT for NU21, respectively, illustrates
a net increment of 8 kt in both the simulations. However, only the vapour increment due to overshoots is 2 kt in REF and
3 kt in NU21, respectively. We associate this contrast of vapour enhancement to the different particle size distribution in the
simulations, consequently suggesting a variation in the sedimentation process. Another fact could be the longest lifetime since
the last overshoot above 17 km in NU21, which is longer than REF. Therefore, ice particles injected in the stratosphere should
have more time to sublimate in NU21 than REF.

In REF, we explain the peak of total water content at 16:22 UT with the last two overshooting events that occurred at 16:15
UT, refer to Fig. 8a and Table 1, injecting a bulk amount of $H_2O$ remaining in the lower stratosphere. We observe two more
events occurring at 16:37 UT, causing a modest enhancement in the total water mass. Subsequently, the last overshooting
event at 16:52 UT is not significant enough to add $H_2O$ to the lower stratosphere. Now, in NU21, the triggering time of the
overshooting events is different than REF, where we observe several peaks in Fig. 8b during 15:00-16:37 UT. Recalling the
results in Table 1, it does not produce as many overshoots as REF during the period of observation, although, represents more
intense overshoots reaching higher than 19 km. Besides, the rise in total $H_2O$ values after a decline at 16:15 UT is possible
because of other new overshoots, overpassing 380 K layer, but not recognised due to the below threshold height of <17 km.





Moreover, we determine the standard amount of hydration for each overshoot, providing both the upper and lower limit by reflecting the two extreme cases on the fate of ice. Such as (1) the upper limit would assume all the remaining ice sublimates in the stratosphere, and (2) the lower limit would indicate all the remaining ice is falling back to the troposphere without sublimating at all. The upper limit is about $\frac{8\,\mathrm{kt}}{6} \approx 1.34\,\mathrm{kt}$ in REF, whereas it is $\frac{8\,\mathrm{kt}}{4} = 2\,\mathrm{kt}$ in NU21. The lower limit of hydration for REF is $\frac{2\,\mathrm{kt}}{6} \approx 333\,\mathrm{t}$, whereas for NU21, it is $\frac{3\,\mathrm{kt}}{4} \approx 750\,\mathrm{t}$. In both the cases during 15:00-17:30 UT, the denominator denotes the
total number of overshooting turrets, denoted by arrows in Fig. 8, and the numerator gives the net amount of WV enhancement. The lower limit is an important point, which is unlikely to be reached because of the very weak fall speed of the small size pristine ice and snow particles.

    Fig. 8 also confers some information on the total amount of ice injected by an individual overshooting plume. For REF at 15:37 UT, we observe $\sim 2\,\mathrm{kt}$ of ice enhancement because of one overshooting plume and later at 16:15 UT, $\sim 11\,\mathrm{kt}$ because
of two more overshoots. The contribution of one overshooting event is thus $\frac{13\,\mathrm{kt}}{3} \approx 4.3\,\mathrm{kt}$ of ice only. Following the identical strategy for NU21 at 15:07 UT, the ice enhancement due to single overshooting event is $\frac{8\,\mathrm{kt}}{2} = 4\,\mathrm{kt}$. Several mesoscale modeling studies (e.g., Liu et al., 2010; Lee et al., 2019) and satellite observations (e.g., Iwasaki et al., 2010; Lelieveld et al., 2007) have already reported regarding this type of total water enhancement due to overshoots in the tropical lower stratosphere. Dauhut et al. (2015) estimate about $2776\,\mathrm{t}$ of WV enhancement, and Lee et al. (2019) estimate a water budget of $869\,\mathrm{t}$. Our
calculation: $\sim 1300\,\mathrm{t}$ in REF and $2000\,\mathrm{t}$ in NU21, ranges between these studies and is of the same order of magnitude. However, this calculation is significantly higher than the estimation of Liu et al. (2010), $\sim 500\,\mathrm{t}$ at maximum, where they use the same version of the BRAMS model to analyse the overshoots occurring in West Africa but is less constrained by observations. On the other hand, Dauhut et al. (2018) provides the estimation of the individual contributions of each overshooting plume hydrating the stratosphere, leading to a lower estimate. However, the method applied to get this estimation is absent. Overall
our estimations on the total H$_2$O enhancement are compatible with most of these studies. They could pave the way for forcing the impact of overshoots in a large-scale computing cost-effective simulation, which cannot resolve overshoots due to coarser horizontal representation.

    To get quantitative information on the mass distribution of five different types of ice hydrometeors within the overshooting plumes constrained within the thin layer of $380\,\mathrm{K}$ to $430\,\mathrm{K}$ isentropes (see Fig.5), we estimate the percentage of each type of
ice particles. It follows in two ways: (1) $\rho_1 = \frac{m_i}{M_i} \times 100$, where $m_i$ corresponds to the mass of a particular type of ice particles $i$ within a layer of $380\,\mathrm{K}$ to $385\,\mathrm{K}$, and $M_i$ corresponds to the mass of the same type of ice particles $i$ within a layer of $380\,\mathrm{K}$ to $430\,\mathrm{K}$; (2) we express them as a percentage of the mass of a given kind of ice particle to the total mass M of ice particles, $M = \sum_{i=1}^{5} M_i$, within a layer of $380\,\mathrm{K}$ to $430\,\mathrm{K}$, namely, $\rho_2 = \frac{M_i}{M} \times 100$. We tabulate the results in Table 2.

    One of the major inferences drawn from Table 2 is the amount of ice injected by various overshooting plume remaining
within a layer of $380\,\mathrm{K}$ to $385\,\mathrm{K}$, $\rho_1$: $\sim 72\%$ in REF and $\sim 65\%$ in NU21. The $\rho_1$ and $\rho_2$ highlight the conclusions of sect. 5.1, i.e., the overshooting plume is essentially comprised of pristine ice, snow, and aggregates, though it can contain a small amount of graupel, present mostly at $380\,\mathrm{K}$ to $385\,\mathrm{K}$, the base of the plume. Furthermore, within $380\,\mathrm{K}$ to $430\,\mathrm{K}$, hail is negligible in the overshooting plume for both the simulations but is always the dominant hydrometeor in the base of the plume, featuring the results of radar observations in Homeyer and Kumjian (2015). We also recognise competition in the growth of





pristine ice over aggregates and graupel concurrently within the plume. Whenever aggregates and graupel are relatively large in mass inside the plume, e.g., 15:37 UT in REF and 15:30 UT in NU21, pristine ice prevails relatively low, and vice-versa, e.g., 16:37 UT in REF and 15:52 UT in NU21. It signifies the existence of the weak vertical velocity, which results in settling back of larger particles. Thus, hail and graupel fall back to the troposphere, allowing further growth of smaller ice particles (see Homeyer and Kumjian, 2015; Qu et al., 2020) in the lower stratosphere within an environment comprising a significant

quantity of supercooled liquid water content. In Table.2, the variations in the quantities of individual ice particles above $380\,\mathrm{K}$ layer between the two simulations are possibly due to the small change in the microphysics adopted to investigate the impact of shape parameter ($\nu$) on producing overshoots. Since the $\nu$ value is higher in NU21, the particle size distribution is more limited than in REF. The particle size distribution resulted from a gamma function becomes narrower as the $\nu$ value increases (see equation 1). Consequently, the lesser variability present in the particle size distribution of NU21 could lead to a more

efficient falling back process of larger ice particles to the troposphere in comparison to REF. Besides, recalling the results from Fig. 8, the longer prevalent behaviour of overshoots above $17\,\mathrm{km}$ in NU21 than REF could lead to higher sublimation of ice in NU21, confirms our observation of the less injection of ice in NU21 to the lower stratosphere but results in more hydration.

## 7 Conclusions

This paper outlines several cloud-resolving simulations of convective overshoots penetrating the lower stratosphere, corre-

sponding to the observational case of 13 March 2012 in the frame of the TRO-Pico field campaign at Bauru, Brazil, using the BRAMS mesoscale model. During this series of overshooting convection events, several plumes reached the stratosphere. The S-Band radar, stationed at Bauru, and the balloon-borne measurements from this campaign allow the validation of these simulation results. Then using these simulations, validated to be realistic when compared against TRO-Pico measurements, we obtain the principal physical characteristics of overshooting plumes.

The main results are as follows.

1. Primarily, overshooting plume reaching the lower stratosphere comprises pristine ice and snow, and to some degree aggregates but only at the base, the $380\,\mathrm{K}$ isentropic level.

2. The cross-section of the overshoots at the $380\,\mathrm{K}$ isentropic level is about $450\,\mathrm{km}^2$ and interestingly, it is close to the mother grid resolution, $20\,\mathrm{km} \times 20\,\mathrm{km}$, at which BRAMS cannot determine explicitly the overshooting convection (see

Behera et al., 2018).

3. Overall, 68% of the entire ice mass prevails within the small layer of $380\,\mathrm{K}$ to $385\,\mathrm{K}$. It further indicates that the rest of the 32% ice (principally pristine ice and snow) progresses further up in the stratosphere. That 32% is expected to stay in the stratosphere and sublimate because of the very modest fall speed of pristine ice and snow at altitudes above $385\,\mathrm{K}$, given the subsaturated conditions to ice therein.

4. For this case study, a single overshooting plume injects about $4.15\,\mathrm{kt}$ of ice above the $380\,\mathrm{K}$ level.

5. After sublimation and (or) sedimentation of the stratospheric ice, the stratospheric WV enhancement due to one over-shooting event is estimated to range between $1.34\,\mathrm{kt}$ to $2\,\mathrm{kt}$ as the upper limit and between $333\,\mathrm{t}$ to $750\,\mathrm{t}$ as the lower limit.

These results can be the framework for developing a scheme to drive the impact of overshooting convection on the strato-
spheric water vapour using a computing cost-effective mesoscale simulation of a too moderate resolution that can not compute the overshoots explicitly. This case will be the subsequent step of current work, providing a road map to upscale the impact of overshooting convection on the stratospheric water vapour at a continental scale.

*Data availability.* All TRO-Pico measurements are publicly available at https://cds-espri.ipsl.upmc.fr/etherTypo/index.php?id=1671&L=1, last access: 11 June 2020. S-band radar data can be provided upon request to EDR. BRAMS model set up is publicly available at http:
//brams.cptec.inpe.br/, last access: 18 June 2021.

*Video supplement.* Two videos are provided for the time series analysis made every $7.5\,\mathrm{min}$ in the supplementary material: one for the modelled cloud tops and corresponding S-band radar echo tops; the second one for the vertical distribution of horizontal cross-section of different hydrometeors within the overshooting plume.

*Author contributions.* EDR and AKB conceptualised the study design, methodology, validation, and analysis. JB provided the support to
run BRAMS in different HPC machines, and EDR provided the resources to achieve the simulations. AKB and EDR wrote the original draft and all authors reviewed the paper. EDR and VM received the funding for this research. SMK provided the FLASH measurements, and MG provided the Pico-SDLA measurements. GH gave the meteorology and interpretation of S-band radar data. All authors have read and agreed to the published version of the manuscript.

*Competing interests.* The authors declare that they have no conflict of interest.

*Acknowledgements.* This study is based on a case observation of the TRO-Pico campaign. TRO-Pico is a French ANR funded project (https:
//anr.fr/Project-ANR-10-BLAN-0609), with collaboration from the IPMET institute in Bauru, State of São Paulo, Brazil. We acknowledge all the technical team of IPMET that helped with balloon launches. The project LEFE (Les Enveloppes Fluides) 'Overshoot à Grande Echelle' also provided funding for this work. Computation resources were granted by CINES under the Genci (Grands Equipements Nationaux de Calcul Intensif), projects: A0010105036 and A0030105036, and by ROMEO of Université de Reims Champagne-Ardennes. In the paper, we
use the information that ice was not detected by LOAC (PI: Jean-Baptiste Renard at LPC2E, CNRS and Université d'Orléans, France), and by COBALD (PI: Frank Gunther Wienhold at Institut für Atmosphäre und Klima, ETH Zürich, Switzerland). Both of them are acknowledged.





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



| End of 7.5 min volume scan (UT) | Altitude ($\pm 0.5$ km) and number of plumes | | | |
| --- | --- | --- | --- | --- |
| | Radar | REF | NU21 | HVR |
| 15:08 | | | 17    2x | |
| 15:15 | | | | |
| 15:22 | | | | 17    1x |
| 15:29 | | | 18    1x | |
| 15:37 | | 17    1x | | |
| 15:46 | | | | 17    2x<br>18    1x |
| 15:53 | | | 19    1x | |
| 16:01 | | | | |
| 16:08 | 18    1x | | | 19    2x |
| 16:15 | | 17    1x<br>18    1x | | |
| 16:22 | 17    1x | | | 17    3x |
| 16:29 | | | | |
| 16:37 | | 18    2x | | 17    1x<br>18    1x |
| 16:46 | 19    1x | | | 18    1x |
| 16:53 | 18    1x<br>19    1x | 17    1x | | |
| 17:01 | | | | |
| 17:08 | 18    2x | | | |
| 17:15 | | | | |
| 17:22 | 18    1x | | | |
| 17:29 | | | | 18    2x |
| 17:37 | 19    1x | 17    1x | | 18    1x |
| 17:46 | | | 17    1x | |
| 17:53 | | 17    1x | | |
| 18:01 | | | | 17    1x |
| 18:08 | | 17    1x | | |
| 18:15 | | | | |
| 18:22 | | | | 17    2x |
| 18:29 | | 17    1x | | |
| 18:37 | | | 17    1x | |

**Table 1.** Ending time of Volume Scan (UT) when one or more overshooting plumes, denoted by multiple of $x$, observed and their corresponding altitudes ($\pm 0.5$ km), tabulated for the S-band radar, REF, NU21, and HVR. The altitude corresponds to the minimum height within a 1 km thick layer.



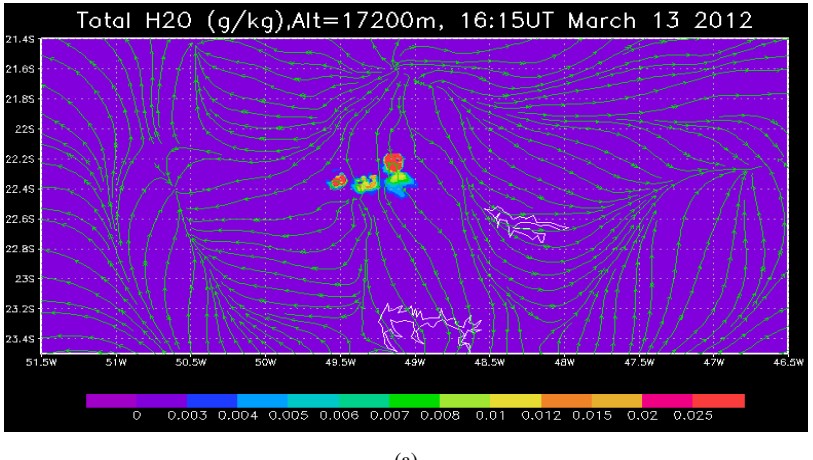

(a)

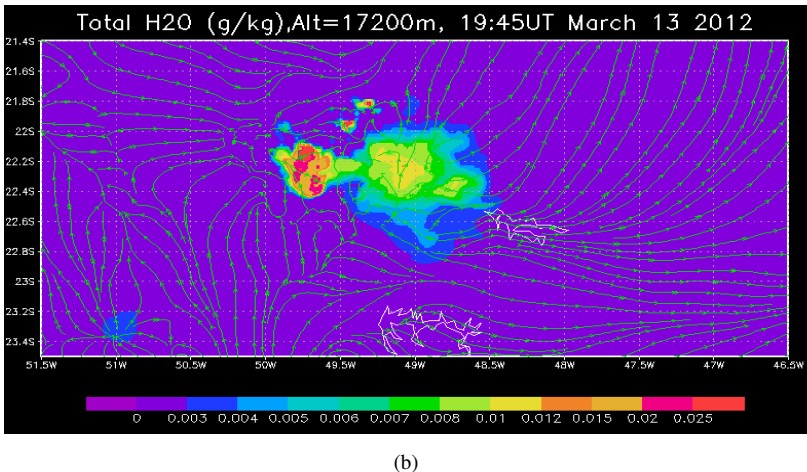

(b)

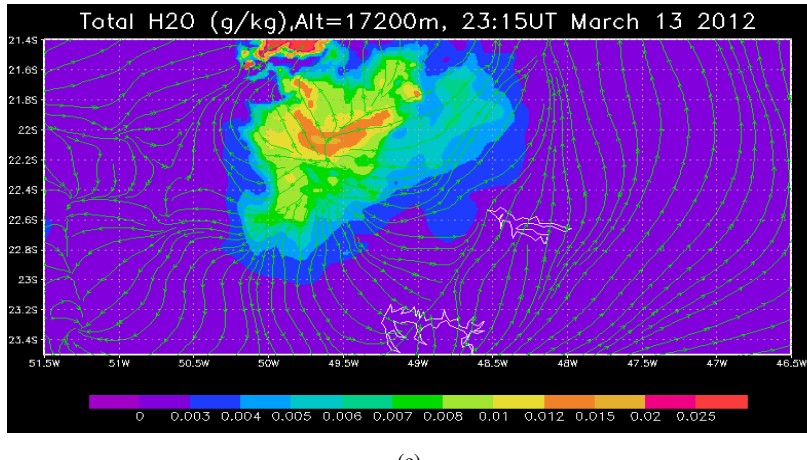

(c)

**Figure 2.** BRAMS simulation: REF total water content, ice + liquid + vapour in $\mathrm{g\,kg^{-1}}$, at 17.2 km altitude at a) 16:15 UT, b) 19:45 UT, and c) 23:15 UT, respectively. The streamlines represent the horizontal wind fields within the domain, a composite of the second and the third grid.



(a) 17.2 km level at 23:15 UT

(b) 17.8 km level at 20:52 UT

**Figure 3.** REF providing total water (ice + liquid + vapour) enhancement at: (a) 17.2 km altitude at 23:15 UT and (b) 17.8 km altitude at 20:52 UT, respectively. The top panel shows the horizontal cross-sections of the vertical grid at these altitudes, depicting the only grid-points when their total water content is higher than the model levels simply above and below in a vertical column. The isolines show the enhanced total water content (ppmv) with respect to the model layer below it. The bottom panel shows the grid-points/pixels' water content confined by the northeast tilting rectangle having the length of 70 km and half-width of 25 km. The red triangle denotes Bauru (0 km); the northeast direction is positive and vice-versa.



(a) 17.2 km level at 23:15 UT

(b) 17.8 km level at 20:52 UT

**Figure 4.** Like Fig. 3 but represents NU21.



**Figure 5.** Vertical distribution of horizontal cross-section of hydrometeors, viz., snow, pristine ice, graupel, and aggregates, within the third grid, spanning over 15 km to 19 km altitude. It is for the ratio of four types of ice hydrometeors against the entire ice content from REF - upper panel, and NU21 - lower panel, shown at 16:15 UT. We skip hail because of its negligible values within the plume.





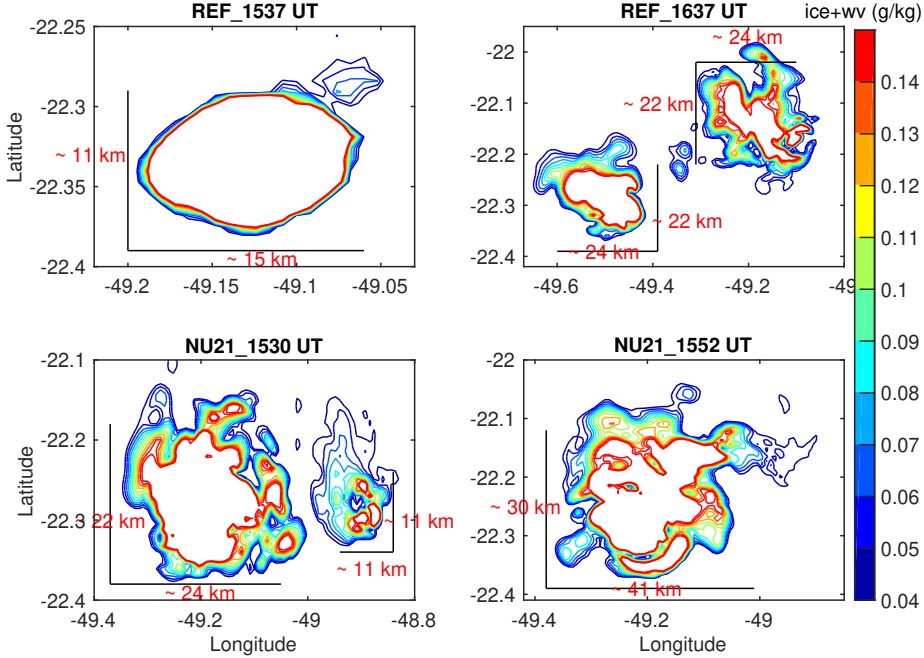

**Figure 6.** Size of the overshooting plumes at $380\,\mathrm{K}$ isentropic level, shown for REF - upper panel: at 15:37 UT and 16:37 UT, and NU21 - lower panel: at 15:30 UT and 15:52 UT, respectively. These times are selected from Table 1. The colour contours show certain levels from $0.04\,\mathrm{g\,kg^{-1}}$ to $0.14\,\mathrm{g\,kg^{-1}}$ of total $H_2O$ content to highlight the outer part of the overshooting plumes. The solid black lines give the approximate range of each figure in $\mathrm{km}$.

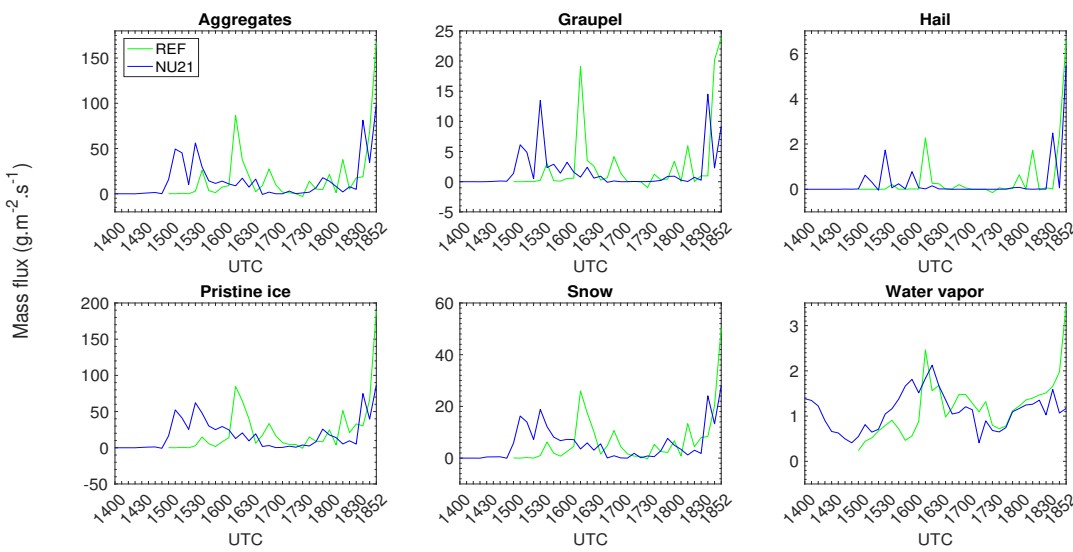

**Figure 7.** Instantaneous domain-average mass-flux rate $(\mathrm{g\,m^{-2}\,s^{-1}})$ of each hydrometeor across the $380\,\mathrm{K}$ isentropic level in the third grid of the simulations, shown for REF - green and NU21 - blue.





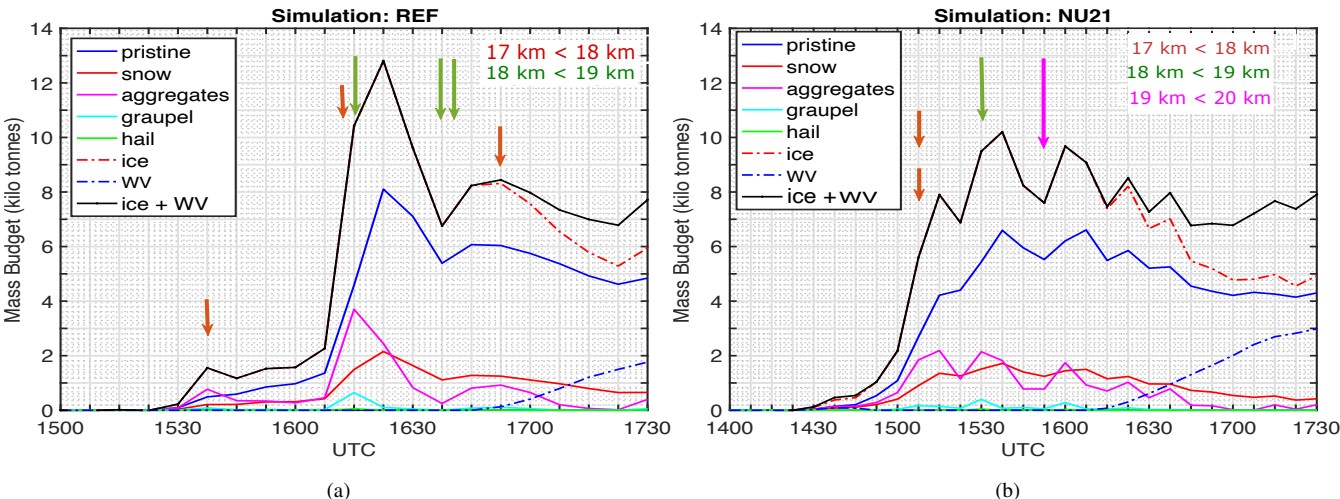

(a)                                                          (b)

**Figure 8.** Mass budget of five different ice hydrometeors including total ice and water vapour within the slice of 380 K to 430 K isentropic levels, shown for (a) REF and (b) NU21. The arrows denote the time of triggering of overshoot, while their colours show the intensity of each event.

| Cases | | Pristine | Snow | Aggregates | Graupel | Hail |
|---|---|---|---|---|---|---|
| 15:37 UT : REF | $\rho_1$ | 69.60 | 68.86 | 70.63 | 73.78 | 80.07 |
| | $\rho_2$ | 31.95 | 13.39 | 48.97 | 5.39 | 0.31 |
| 16:37 UT : REF | $\rho_1$ | 61.78 | 56.50 | 71.48 | 78.23 | 83.92 |
| | $\rho_2$ | 79.71 | 16.45 | 3.70 | 0.14 | $2.63 \times 10^{-5}$ |
| 15:30 UT : NU21 | $\rho_1$ | 59.26 | 55.80 | 59.97 | 62.90 | 70.53 |
| | $\rho_2$ | 57.03 | 15.84 | 22.52 | 4.17 | 0.44 |
| 15:52 UT : NU21 | $\rho_1$ | 62.06 | 55.60 | 63.96 | 70.34 | 79.43 |
| | $\rho_2$ | 72.64 | 16.32 | 10.29 | 0.73 | $8.0 \times 10^{-5}$ |

**Table 2.** Mass (%) of individual ice hydrometeors within 380 K to 385 K isentropic layer ($\rho_1$) and 380 K to 430 K isentropic layer ($\rho_2$), respectively, with respect to its total ice mass within 380 K to 430 K isentropic layer. Results are tabulated for four different cases: first two rows for REF and the rest for NU21.