# Peer review of "On the cross-tropopause transport of water by tropical convective overshoots: a mesoscale modelling study constrained by in situ observations during TRO-Pico field campaign in Brazil"

_Atmospheric Chemistry and Physics, 2021_

## Referee Comment (RC1)

Case 1

z

380K ----------------------------------------

cloud

x

Case 2

z

380K ------------------------------------

cloud

x

Case 3

z

380K - - - - - - - - - - - - - - - - - - -

cloud

x

a few minutes later

z

380K - - - - - - - - - - - - - - - - -

cloud

x

---

## Author Comment (AC2)

We greatly appreciate the reviewer's helpful and informative suggestions, which prompted us to revise our article. We appreciate your opinion of our English skills as well. To be honest, we are unable to undertake additional sensitivity tests to evaluate the differences between REF and NU21 runs due to the time and computational cost of these simulations, which necessitates additional time to analyse. Otherwise, we reran the simulations with a 30-second time resolution, which is critical for answering several of reviewer-1's questions. Here is where we record our point-by-point response to the reviewer's comments. The original comments of the reviewer are listed. Italic and boldface typefaces are utilised in the typesets. Each remark is met with a response from us. The phrase "adjusted" is always included in the response when any changes are made to the original version of the manuscript. The line numbers, page numbers, figure numbers, and table numbers refer to the original version of the document unless otherwise stated. The corrected version of the manuscript is also attached.

**1 Major Comments**

1. *In regards to the nu21 simulation, can you describe what the change in the shape parameter from 2 to 2.1 actually means? The text says: A larger $\nu$ implies a larger modal diameter with a narrower distribution width. Please give quantitative values at least in regards to model diameter in the text and a sense of the distribution width?*

   adjusted - Please see Fig. 1 in the study by [Walko et al., 1995]. The paragraphs corresponding to line 155, on the other hand, have been revised to include more information on the shape parameter and the objective of employing NU21 simulation.

   The text is now: "In Eq. 1, $f_{gam}$ denotes the probability density function for the modified gamma distribution of hydrometeors with a diameter of D, as obtained from [Walko et al., 1995]. $\Gamma(\nu)$ is the normalisation constant, and $D_n$ is the characteristic diameter of the modified gamma distribution. A bigger $\nu$ indicates a narrower distribution width and a larger modal diameter. As a result, the proportion of smaller and bigger hydrometeors in the distribution is modulated. The size distribution of hydrometers would be more peaked as the modal diameter increased."

   Nonetheless, we have included a figure (#1#) that compares REF and NU21 mean mass diameter variability in altitude around overshoots for pristine ice, snow, and aggregates. As expected, NU21 has a slightly larger mean mass diameter than REF, at least for pristine ice, which is the first ice hydrometeor to be formed by freezing. Other hydrometeors have a more complicated conclusion because they arise from pre-existing ice particles. We should also emphasise that this comparison is essentially indicative; because the cells are not identical, they are not strictly comparable one to one.

[Figure]

| (a) Pristine | (b) Snow | (c) Aggregates |

**1# Vertical profiles of mean mass diameters in the vicinity of overshoots for REF and NU21, respectively. The black lines are for REF at 16:15 UT, 22.0°S, 49.18°E. The green lines are for NU21 at 15:45 UT, 22.0°S, 49.2°E (almost the same position but not the same time). Those times correspond to the ones in Fig. 1 of the submitted paper. The positions correspond to the positions of the maximum overshoot in each case.**

2. *Why does the high vertical resolution simulation produce unrealistic results (too much convection)*

adjusted - We are surprised to see such deep convection behaviour in HVR run. We could not help ourselves when we observed the animation of cloud temporal progression compared to the S-band radar and the other two simulations. Exaggerated values in HVR could be due to unusually excessive convection, which includes extremely high upward velocity. Please see the attached figures on vertical wind (w) profiles (#2#) and CAPE maps (#3#) in the vicinity of overshoots, related to the Fig.1 in the submitted paper. Aside from that, we have expanded on other plausible explanations for such behaviour, such as the CFL limit and the need for Von Neumann stability testing in BRAMS. This is the first one we address because simulating HVR was somewhat difficult. Due to unexpected failures with no particular hint or reason for failure during the run, we had to make several history restarts to complete the simulation. The paragraph that corresponds to lines 261-269 has been changed.

The text is now: "To further understand the situation, one can expect HVR to determine more reliable dynamics across the tropical tropopause than REF and NU21, respectively. Contrary to expectations, it tends to intensify massive deep convection activity. A plausible fact to explain such behaviour in HVR is the ratio between vertical and horizontal grid points, which overestimates vertical motions due to grid cell saturation [Homeyer et al., 2014, Homeyer, 2015]. It might be the model's Courant–Friedrichs–Levy (CFL) limit, which in finite-difference simulation techniques constrains the relationship between infinitesimal increases in space grid points and infinitesimal time step increments. In the BRAMS model, the von Neumann stability assessment [Deriaz and Haldenwang, 2020] is necessary for the transport equations related to convection. Aside from that, Eulerian model simulations of high vertical resolution, high-frequency wave motions, such as inertial-gravity waves [e.g., Staquet, 2004, Young, 2021], can be overdetermined. As a result, they can exaggerate cloud microphysics [Aligo et al., 2009] and cause erroneous cloud conditions near the TTL [Jensen and Pfister, 2004]. Therefore, we leave HVR out of the next sections to describe the

details, and we do not look at this simulation's water budget in the lower stratosphere."

[Figure]

(a) REF at 16:15 UT    (b) NU21 at 15:45 UT    (c) HVR at 15:45 UT

**2# REF (a), NU21 (b), and HVR (c) vertical wind profiles (w) in the vicinity of overshoots, respectively. The time and location of these vertical profiles are determined in accordance with the submitted paper's Fig. 1. It is worth noting that HVR has a surprising high velocity in the 16 km to 18 km altitude region, which is essentially non-existent in REF and NU21.**

[Figure]

(a) REF at 16:15 UT    (b) NU21 at 15:45 UT    (c) HVR at 15:45 UT

**3# CAPE maps ($J\,kg^{-1}$) for REF (a), NU21 (b), and HVR (c), respectively. According to the submitted paper's Fig. 1, the time these maps are determined. It is worth noting that HVR has a rather high CAPE value, whilst REF and NU21 have almost none.**

3. *In Figure 7, the last panel is a water vapor flux. That is not a hydrometeor (as indicated by the caption). Is it just calculated based on vertical velocity and gaseous $H_2O$? The caption should indicate what is plotted.*

adjusted - Yes, the flux rates are calculated by projecting the vertical velocity against the normal to the isentropic level of 380 K. The isentropic level deformation caused by deep convection is taken into account. To do so, we compare the heights of surrounding gridpoints in vertical grid coordinates in meters to determine the slope of the 380 K surface. It is worth noting that the $\delta x$ and $\delta y$ in this example is 800 m, which is the third grid's resolution. We take vertical velocity and the accompanying gaseous $H_2O$ into account when calculating water vapour flux rates.

The figure caption is now: "The instantaneous domain-average mass-flux rate (g m$^{-2}$ s$^{-1}$) of each hydrometeor and water vapour is illustrated in the third grid of the simulations for REF (green) and NU21 (blue). The cosine component of the vertical velocity with respect to the horizontal is used to determine the upward flux rate, which takes into account the slope at the 380 K level due to deep convection."

4. *This figure confuses me; it could be made more understandable with a more detailed figure caption. Is the black line just the sum of the 5 hydrometeors + water vapor? What is ice? Is that just the sum of the 5 hydrometeors? And, what does it mean that the colors show the intensity of the event? Aren't you showing the altitude of the event as opposed to intensity? And, why does the plot start off with no water vapor in the region of interest? The caption (as well as the text) should say that this is a plot with respect to the unperturbed state. And, why doesn't the run go out any further in time? (as that is what is needed to assess how reversible the flux into the stratosphere was.)*

adjusted - Snow, pristine ice, graupel, aggregates, and hail have all been mentioned specifically in the figure description. The combination of these five hydrometeors is referred to as ice. Cloud top height replaces the perplexing word intensity. The cloud top height is shown by the colours and lengths of the arrows, while their placements represent the times of overshoot. Lines 430-435 have been updated in the paragraph. The unperturbed state was noted in the caption, and it is also referenced in line 430 of the text.

The caption is now: "Water mass budget (ice and water vapour) for (a) REF and for (b) NU21 in the third grid between the 380 K to 430 K isentropic levels. The ice budget contribution includes the five ice hydrometeors (pristine ice + snow + aggregates + graupel + hail). The colour and length of the arrows indicate the cloud top altitude of each occurrence, with the smallest arrows (brown) referring to cloud top heights of 17 km to 18 km, the intermediate-sized arrows (green) relating to cloud top heights of 18 km to 19 km, and the largest arrows (magenta) corresponding to cloud top heights greater than 19 km."

Of course, the runs continue past 17:30 UT. Please see the animation on cloud top heights with S-band radar in the Supplementary Materials. When there is no convective activity in the model, we consider it as an unperturbed condition in the runs. The water vapour profile reaches a near plateau profile near the end of the mass estimation period, i.e., 17:30 UT in both runs, while the ice (sum of 5 hydrometeors) profile shows a descending trend - no more deep convection, indicating that ice is sublimating or possibly falling back into the troposphere and besides we observe a rise in water vapour budget. If we look at the simulations over a longer period, we can see that more overshoots are induced. The corresponding flux rates are shown in Fig. 7. At about 17:30 UT, both simulations reveal almost no entrance of hydrometeors and water vapour at the 380 K level. In Fig. 8, we plan to calculate the mass budget associated with individual overshoots. In regard to your point about reversible fluxes, we include a range of values from the near plateau at the end of the observation period to allow for the possibility of ice falling down to the troposphere rather than sublimating locally. However, it is not possible to directly compute the sedimentation flux through the tropopause in the model since it depends not only on the vertical wind velocity but also on the fall speed of each hydrometeor. It would have required rewriting BRAMS' microphysical framework, introducing extra scalar variables, and rerunning all of the simulations reported here, which was not feasible given the time constraints.

5. *Conclusion 3 says: "It further indicates that the rest of the 32% ice (principally pristine ice and snow) progresses further up in the stratosphere." Was the model run long enough*

*to verify this?*

adjusted - This estimate is based on simulations till 17:30 UT. The complete simulations, on the other hand, ran until March 14th at 12:00 UT. You may refer to line 184 for further information. We have until March 13th, 22:00 UT to validate our models using the S-band radar. Also, at 18:53:38 LT (UT-3 hours), there were no more echoes on the radar screen, indicating that all convective activity had ceased within a 100 km radius. Nevertheless, the simulations appear to be active in producing deep convection until 18:45 UT, which is not the case in radar images in the nearby area of Bauru, according to the animation of the radar versus model comparison of cloud tops. Moreover, Table 1 should be consulted. Based on the existing analysis, it is difficult to say whether snow and pristine ice still exist in the dome's higher layers. However, because the fall speed of these two hydrometeors is so slow, it has been ascertained that a situation in which a considerable amount of them fall back to the troposphere before sublimating is extremely unlikely. Otherwise, we have rewritten the third point in the conclusion to avoid any misunderstandings. We place emphasis on the fall speed of pristine ice and snow, implying that they have a very slight possibility of falling back to the troposphere.

The text is now: "#3. Within the modest layer of 380 K to 385 K, 68% of the overall ice mass exists. It also suggests that the remaining 32% of ice (mostly pristine ice and snow) moves higher in the stratosphere. Because of the very slow fall speed at altitudes above 385 K and the subsaturated conditions with respect to ice, that 32%, which is pristine ice and snow, is anticipated to stay in the stratosphere and sublimate."

6. *Conclusion 4 says "For this case study, a single overshooting plume injects about 4.15 kt of ice above the 380K level." Where does this value of 4.15 kt come from? Is this an average of the two model simulations? If I then apply Conclusion 3, which seems to say that 32% is irreversibly injected into the stratosphere, then I get close to the 1.34 kt noted as the minimum in the upper limit range noted.*

adjusted - Yes, the ice injection values from the two scenarios are averaged. We have clarified it now. Also, thank you for your observation about the ice budget; it is precisely how you see it. Points 3, 4, and 5 have been modified in the conclusion.

The text is now: "#4. A single overshooting plume injects around 4.3 kt of ice in REF and 4.0 kt of ice in NU21 over the 380 K level in this given scenario in Bauru, with NU21 injecting slightly less ice than REF as expected. #5. The stratospheric WV enhancement due to one overshooting event is estimated to range between 1.34 kt to 2 kt as the upper limit and 0.34 kt to 0.75 kt as the lower limit after sublimation and (or) sedimentation of the stratospheric ice. If we consider complete sublimation of ice, as in REF, it confirms our estimate that the 32% of 4.3 kt of ice irreversibly traveling further up to the stratosphere results in the stratosphere having the lowest hydration in the upper limit range."

7. *In regards to the calculation of the lower limit, is that just making an estimate based on what is water vapor at the end of the simulation? And, another question, why is the ref simulation 2.5 hours, and the nu21 simulation 3.5 hours? Isn't the amount of water vapor at the simulation end going to be a function of how long the simulation actually was?*

– Yes, our lower limit for $H_2O$ injection into the stratosphere is the scenario in which all of the ice particles have sedimented, so the remaining $H_2O$ in vapour form provides the lower

limit. And this is not the total vapour until the end of the simulation; it is still 17:30 UT, as previously discussed. Both runs reach a plateau above the 380 K level around 17:30 UT. The runs, however, end at 12:00 UT on March 14th. Your question about the observation period in REF and NU21 is perfectly valid, and we have thought about it as well. To avoid such uncertainties, we consider the simulations from a time point where both are in the same stage, i.e., the unperturbed state with respect to convection activity. REF needs only 2.5 hours to achieve a stable state in terms of water vapour budget above a 380 K isentropic threshold, whereas NU21 takes 3.5 hours. Finally, your assessment of the duration of the run and water vapour enhancement is correct; NU21 accumulates approximately 3 kt of water vapour, which is relatively higher than REF. However, the criteria for estimating the mass budget for both REF and NU21 runs are the same.

8. ***This seems to be results for a specific meteorological event, and not necessarily extractable to a general case, so I question the final conclusion that this study provides "a road map to upscale the impact of overshooting convection on the stratospheric water vapour at a continental scale."***

adjusted - Thank you for pointing this out. These findings do not imply that this is a common occurrence. However, more case studies like this one are required to develop a global picture of overshoots utilising cloud-resolving simulations from various locations. This reported study is the consequence of many overshooting domes reaching different elevations above the 380 K isentropic layer, despite the fact that it is a single-case study. In that way, it also accounts for the variation under hydration induced by overshoots of varying intensity - the cloud top height, even when they occur in similar circumstances (e.g., stratospheric humidity). Otherwise, the lines between 495 and 515's corresponding paragraphs have been changed.

The text is now: "This paper describes several cloud-resolving simulations of convective overshoots penetrating the lower stratosphere using the BRAMS mesoscale model, corresponding to an observed case on March 13, 2012, during the TRO-Pico field campaign in Bauru, Brazil. During this series of overshooting convection events, several plumes reached the stratosphere. As a result, it accounts for the hydration heterogeneity produced by overshoots of variable intensity, even when they occur under similar circumstances (e.g., stratospheric humidity). The S-Band radar stationed at Bauru, as well as the balloon-borne measurements from this campaign, allow the simulation results to be validated. These simulations, which have been validated as realistic when compared to TRO-Pico measurements, are then used to obtain the main physical characteristics of overshooting plumes.

These data can be utilised to develop a nudging method that quantifies the influence of overshooting convection on the stratospheric water vapour using a low-cost, large-scale simulation. Though the findings are limited to a case study in Brazil and may not be generalisable, more of similar case studies should be conducted in order to gain a better knowledge of the events, and this work is in keeping with that goal. This instance would be the next stage in the current research, offering a road map for extending the impact of overshooting convection on stratospheric water vapour on a continental (Brazilian) scale."

**References**

Robert L Walko, Wr R Cotton, MP Meyers, and JY Harrington. New rams cloud microphysics parameterization part i: the single-moment scheme. *Atmospheric Research*, 38(1):29–62, 1995.

Cameron R Homeyer, Laura L Pan, and Mary C Barth. Transport from convective overshooting of the extratropical tropopause and the role of large-scale lower stratosphere stability. *Journal of Geophysical Research: Atmospheres*, 119(5):2220–2240, 2014.

Cameron R Homeyer. Numerical simulations of extratropical tropopause-penetrating convection: Sensitivities to grid resolution. *Journal of Geophysical Research: Atmospheres*, 120(14):7174–7188, 2015.

Erwan Deriaz and Pierre Haldenwang. Non-linear cfl conditions issued from the von neumann stability analysis for the transport equation. *Journal of Scientific Computing*, 85(1):1–17, 2020.

Chantal Staquet. Gravity and inertia-gravity internal waves: Breaking processes and induced mixing. *Surveys in geophysics*, 25(3-4):281–314, 2004.

William R Young. Inertia-gravity waves and geostrophic turbulence. *Journal of Fluid Mechanics*, 920, 2021.

Eric A Aligo, William A Gallus, and Moti Segal. On the impact of WRF model vertical grid resolution on Midwest summer rainfall forecasts. *Weather and forecasting*, 24(2):575–594, 2009.

Eric Jensen and Leonhard Pfister. Transport and freeze-drying in the tropical tropopause layer. *Journal of Geophysical Research: Atmospheres*, 109(D2), 2004.

---

## Author Comment (AC3)

* * *
**Authors' comments on the first revision of the manuscript: "On the cross-tropopause transport of water by tropical convective overshoots: a mesoscale modelling study constrained by in situ observations during TRO-Pico field campaign in Brazil", Reviewer-3**
* * *
We greatly appreciate the reviewer's helpful and informative suggestions, which prompted us to revise our article. Thank you also for recognising the long-term study efforts to assess the influence of overshoots on the lower stratosphere's water budget. We make every effort to improve our style following your suggestions. This document contains a point-by-point answer to your remarks. There is a list of the original remarks. The typesets feature italic and boldface typefaces. Each remark is met with a response from us. The phrase "adjusted" is always included in the response when a change is made to the original version of the manuscript. The line numbers, page numbers, figure numbers, and table numbers refer to the original version of the document unless otherwise stated. The corrected version of the manuscript is also attached.

**1 Major Comments**

1. *l65-69: this paragraph is extremely important as it sets the long-term strategy followed by this research group, involving field campaigns and numerical simulations at different resolutions. This is therefore a key pargraph. It would be good to make it a bit more precise and expand a bit. What is considered nececssary as "fine-scale" simulations? In upscaling or generalizing results from fine-scale simulations to a larger scale, with parameterizations in mind, how do the authors suggest to tackle the issue of representativity? What do they expect the key variables from the large-scale state and circulation to be? What are candidates (for the large-scale variables that would, in a parameterization, influence the occurence or not of overshoots)? CAPE near the surface? The stability near the tropopause? How do these questions influence the design of case studies?*

   adjusted - To avoid any misunderstanding, this paragraph has been revised. We are talking about cloud-resolving simulations when we say "fine-scale" simulations. CAPE at the surface has a big impact on convection, and grid resolution has a big influence on it. Our research, on the other hand, is focused on the water vapour budget in cloud-resolving simulations where CAPE is high enough, and cloud-scale dynamics and TTL dynamics are good enough to induce overshoots. This calculation of the overshoot impact on the stratospheric water budget can then be utilised to feed a nudging strategy in a large-scale (Brazilian scale) simulation. Certainly, the estimate provided in the submitted paper does not represent the full range of hydration overshoots, and more results from other studies, including those conducted by our group, are required. However, in the simulations described in this paper, numerous overshooting cells exist, and the quantities presented here represent the average contributions of these several overshooting cells. We have chosen to share only broad information about how the nudging system should be since we do not want to reveal detailed indicators of what our nudging scheme (under development) will be until it is ready for publishing. In addition to what we want to undertake in our group in the near future, our estimates could be employed in a similar approach to superparameterization as detailed in Grabowski [2001], Khairoutdinov and Randall [2001], Khairoutdinov

et al. [2005]. The goal of superparameterisation is to explicitly include a convection parameterisation by a cloud-resolving model at each sub-grid scale in the GCM, integrating the local-scale aspects of convection in terms of the diurnal cycle and strength of convection that are typically unresolved by convective parameterisations in GCMs.

The updated paragraph is now: "Another potential strategy is to upscale stratospheric overshooting effects by forcing them into a large-scale simulation, where the overshoots are explicitly resolved in cloud-resolving numerical simulations. However, cloud-resolving simulation studies of several cases must be conducted before proceeding with this phase. The combined study of results corroborated by observations would encourage a stratospheric overshoot nudging strategy in a larger-scale or Brazilian size simulation. Furthermore, utilising the superparameterization method [Grabowski, 2001, Khairoutdinov and Randall, 2001, Khairoutdinov et al., 2005], explicitly adding cloud-resolving simulation in each grid or sub-grid point of a general circulation model (GCM) simulation or sub-GCM simulation to consolidate the local-scale aspects such as the diurnal cycle and convection strength [e.g., Khairoutdinov and Randall, 2006] would provide information on the influence of overshoots at a large scale. The goal of this research is to learn more about cloud-resolving simulations."

2. *l87-93: for readers not very familiar with balloon technology, it would be worthwhile to explain a bit more (what are the different balloons inflated with, which are closed, which are open, what are typical ascent rates, flight durations, maximum altitudes? What are the payloads for different balloons, what are advantages and disadvantages?) and include references for interested readers.*

adjusted - To cover the gaps you pointed out, we amended the paragraph concerning lines 87–93 with more informative information about balloon-borne data pertinent to this case study. We have also included references to a resource that goes through the basics of large-scale balloon campaigns and the strategies that go along with them.

The updated paragraph is now:"TRO-Pico is a French initiative based on a small balloon campaign in Bauru (22.36°S, 49.03°W), State of São Paulo, Brazil, and funded by the Agence Nationale de la Recherche (ANR). Its purpose is to study the stratospheric water vapour entry in the tropics at different spatial and time scales. In particular, TRO-Pico main's goal is to better quantify the role of overshooting convection at a local scale in order to better quantify its role at a larger scale with respect to other processes. It took place in March 2012 for the first intensive observation period (IOP) and from November 2012 to March 2013, with regular soundings including a second IOP in January and February 2013. The case under investigation in this paper is part of the first IOP while Behera et al. [2018] investigated the November 2012 to March 2013 TRO-Pico period. Several light-weight devices were used in this campaign, including the Pico-SDLA, which weighs 8 kg, the FLASH-B, which weighs 1 kg, and the COBALD, which weighs 1.3 kg. Hydrogen/helium-inflated Raven Aerostar zero-pressure plastic (open) balloons with volumes of 500 $m^3$ and 1500 $m^3$, as well as 1.2 kg Totex rubber balloons that were somewhat larger than conventional radiosonde balloons, were used. The TRO-Pico campaign provided measurements of $CO_2$, $CH_4$, $O_3$, and $NO_2$ using a large set of equipment. On the other hand, WV and particle measurements were the campaign's main sampling. Only the Pico-SDLA and FLASH-B WV measuring devices, along with the LOAC and COBALD particle measurement equipment, were flown on March 13, 2012. The balloons collected data with a vertical resolution of approximately 20 m. Readers interested in balloon-borne measurement technology may

read Vernier et al. [2018] and Pommereau et al. [2011], as well as the references in those papers, which are based on large balloon campaigns, BATAL and HIBISCUS, respectively."

3. *A major concern is that the authors have only very few simulations to explore the uncertainty on the impacts of overshoots. This is understandable these simulations are costly, and time-consuming to set up, store and analyze. Nonetheless, the simulations that have been made essantially sample the uncertainty due to the model setup. Another set of simulations that would be of interest would be similar simulations (same model setup as the reference simulation for example), but with variations of the large-scale conditions (artificial modifications of, say, lower level humidity, and/or mid-tropospheric humidity, and/or upper tropospheric stability...) to explore the sensitivity of the overshoots to these environmental factors. In the long-term strategy to guide parameterizations, the influence and relative importance of different environmental factors are crucial to estimate, at least qualitatively. It is not reasonable to expect new simulations to be carried out, but such considerations should be explained in a discussion or when sketching perspectives.*

– Thank you for bringing up the analysis time as well as the challenges and limitations that this type of numerical study may entail. To examine the variability of the impact of overshooting convection on the lower WV stratosphere and to aid in the development of a parameterisation, some sensitivity tests adjusting large-scale factors, such as humidity, may be of interest. However, our strategy is different. Our goal is to reproduce observed overshoots as realistically as possible so that we may be confident in our water budget quantifications. More than being the only simulations used to establish a parameterisation, such quantifications are utilised to feed nudging schemes (at least in our group). To be realistic, we choose to follow the ECMWF (operational analyses) initialisation and nudging at the boundary conditions, which are fed by observations via data assimilation.

Nevertheless, the cloud-resolving simulations of Hassim and Lane [2010] assisted to understand the unpredictability of overshoot hydration or dehydration for two instances with very different humidity in the TTL for the sensitivity test altering humidity. In a parameterisation scheme, the lower stratospheric humidity parameter is indeed highly essential. The nudging strategy developed by our group accounts for lower stratospheric humidity by allowing injected ice to sublimate as a function of local humidity.

4. *For the validation, section 4.2, why is the focus so much on the local values? The vertical profiles in different location should be explored? Do simulated vertical profiles, in some places, reproduce the main features of the vertical profiles from balloon measurements?*

– We are focusing on local water vapour enhancements as a result of overshoots because overshoots are rather local. Figs. 3 and 4 in the submitted paper illustrate possible predicted local water vapour enhancements in Bauru at 17.2 km and 17.8 km levels, respectively, where the balloons caught two stratospheric wet air parcels arriving from overshoots during the TRO-Pico campaign. At these heights, the horizontal cross section of the model grid reveals multiple regions of water enhancement. The water vapour enhancement observed by the balloon may potentially be one of many grid points. Finding vertical water vapour profiles in Bauru that match the balloon profiles could be a time-consuming process. Normally, the model's vertical profiles are just that: vertical profiles, whereas balloons fly both vertically and slightly horizontally depending on wind flow, and each balloon delivers data every 20 m vertically, whereas the model is coarse in this regard. We should not expect a typical balloon-like profile. The simulated overshoots also have a little

temporal and spatial shift, as seen in Fig. 1 in the submitted paper. However, for the purpose of simplicity and context, we have extracted some vertical profiles (#1#) based on the local enhancements in Figs. 3 and 4 of the submitted paper, respectively.

[Figure]

(a) REF run at 23:15 UT          (b) NU21 run at 23:15 UT

**1# REF (a) and NU21 (b) vertical total water (ice + liquid+ vapour) profiles in the vicinity of overshoots. The grey colours represent the standard variation of total water content at corresponding heights, based on various grid-points somewhat northeast of Bauru as shown in Figs. 3a and 4a of the submitted paper. The mean value is represented by the red profile.**

**2   Minor Comments**

1. *l5 meteorological model – > climate model (the impact is for climate rather than weather forecasting)*
   adjusted -  The sentence has been rewritten.  Please see the answers below to the next

   suggestion.

2. *l4-6: the sentence is a bit odd in the sense that it suggests three scales: local scale (cloud-resolving model), intermediate scale (mesoscale modelling?) and the global scale (climate models). What is meant exactly for the intermediate scale is not clear.*

   adjusted -  To eliminate any confusion, we have rewritten this text:"Nevertheless, one debatable topic persists regarding the global impact of this event with respect to the temperature driven dehydration of air parcels entering the stratosphere. As a first step, it is critical to quantify their role at a cloud-resolving scale before assessing their impact on a large-scale in a climate model. It would lead to a nudging scheme for large-scale simulation of overshoots."

3. **l9 numerical simulations depend on ...**

   adjusted - This has now been fixed.

4. **l19: '... could establish a forcing scheme...' – > 'could inform the development of / provide guidance for ...'**

   adjusted - The sentence has been rewritten:"In a large-scale simulation, these findings could provide guidance for a nudging scheme of overshooting hydration or dehydration."

5. **l23: is exhibited extensively to be a part – > is known to play an important role in?**

   adjusted - The sentence has been rewritten:"Water vapour (WV) concentrations in the stratosphere impact both chemistry [Shindell et al., 1999, Shindell, 2001, Herman et al., 2002] and Earth's radiative balance [Forster and Shine, 2002]."

6. **l24: was also an element in the formation of polar stratospheric clouds**

   adjusted - The sentence has been rewritten:" It also contributes to the formation of polar stratospheric clouds [Toon et al., 1990, Hervig et al., 1997]."

7. **l29: 'the supercooled temperature field': there is a confusion here. The temperature field is a well-defined physical field. Supercooled water droplets are a thermodynamic phenomenon concerning water.**

   adjusted - The sentence has been rewritten:"In the first order, the very cold temperature field across the tropical tropopause layer (TTL) constrains the abundance of WV in the stratosphere [Holton and Gettelman, 2001, Randel et al., 2001]."

8. **l29: 'drives the abundance' – > 'constrains the abundance'? 'determines...' ?**

   adjusted - This suggestion has been carried out.

9. **l32: beyond the level of positive ... – > above the level of zero radiative heating?**

   adjusted - This suggestion has been carried out:"Inside, above the level of zero radiative heating, air masses progressively ascend and get dehydrated due to solid condensation or sedimentation of ice particles, a process known as the cold-trap mechanism [Sherwood and Dessler, 2000]."

10. **l33: known as the cold-trap ...**

    adjusted - This suggestion has been carried out.

11. **l34-35: It is never certain if such modelling studies 'explain' the abundance of water vapor ... perhaps it is better to write: 'These trajectory studies have found agreement with ...'**

    adjusted - This suggestion has been carried out:"The first trajectory studies by Fueglistaler et al. [2005], James et al. [2008], which ignored the contribution of deep convection in the

TTL, show agreement with the abundance and variability of WV in the tropical tropopause as measured by satellite-borne sensors, confirming the cold-trap as the principal mechanism dominating WV entry into the tropics."

12. *l35: the text should mention that the studies considered here are just the first studies; the reader is otherwise surprised not to find certain more recent studies, which in fact are commented later in the text*

    adjusted - This suggestion has been carried out.

13. *l38: 'conclude' – > show? demonstrate?*

    adjusted - The sentence has been rewritten:"Nonetheless, open-ended debates over the trend of stratospheric WV [Oltmans et al., 2000, Rosenlof et al., 2001, Randel et al., 2006, Scherer et al., 2008] and tropopause temperature [Seidel and Randel, 2006] in the 1990s and 2000s demonstrate that additional factors may be at play in the processes that determine WV entering the stratosphere [Randel and Jensen, 2013]."

14. *l39: 'the processes of WV entering into the stratosphere' – > 'the processes determining the WV entering into ...'*

    adjusted - This suggestion has been carried out.

15. *l42: 'Recently many case studies'*

    adjusted - This suggestion has been carried out:"Recently many case studies, both based on modelling [e.g., Chaboureau et al., 2007, Grosvenor et al., 2007, Chemel et al., 2009, Liu et al., 2010, Dauhut et al., 2015] and observations [e.g., Corti et al., 2008, Khaykin et al., 2009, Iwasaki et al., 2012, Sargent et al., 2014, Khaykin et al., 2016, Jensen et al., 2020], have validated the hydration effect of stratospheric overshoots at local scales in the tropical belt."

16. *l52: 'studies report' – > 'studies suggest'? bring evidence...?*

    adjusted - This suggestion has been carried out:"In recent years, studies suggest that deep convection reaching the tropopause may influence the stratospheric WV budget on a large scale"

17. *l53: 'at a large scale' – > 'on a large scale'?*

    adjusted - This suggestion has been carried out.

18. *l63: 'no studies can' – > 'it has not been possible to ...'?*

    adjusted - The sentence has been rewritten:"However, the critical impact of stratospheric overshoots on the global distribution of WV has so far proven difficult to estimate."

19. *l70: observational – > observed*

    adjusted - This suggestion has been carried out:"Here, we perform three simulations of an observed case of stratospheric overshoots using the BRAMS (Brazilian version of RAMS) mesoscale model."

20. *l72: a range of estimations – > a range of estimates*

    adjusted - This suggestion has been carried out:"It produces a range of estimates on the ice injection into the stratosphere and the remaining water after the sublimation."

21. *l72: the remaining 'water'?*

    adjusted - This suggestion has been carried out.

22. *l85: 'It' : needs to be explained, too abrupt as it is*

    adjusted - This line and the paragraph it refers to have been updated:"TRO-Pico is a French initiative based on a small balloon campaign in Bauru (22.36°S, 49.03°W), State of São Paulo, Brazil, and funded by the Agence Nationale de la Recherche (ANR). Its purpose is to study the stratospheric water vapour entry in the tropics at different spatial and time scales. In particular, TRO-Pico main's goal is to better quantify the role of overshooting convection at a local scale in order to better quantify its role at a larger scale with respect to other processes."

23. *l87: equipped with – > based on ?*

    adjusted - This suggestion has been carried out.

24. *l92: only the WV measuring instruments were flown: Pico-SDLA...*

    adjusted - The sentence has been rewritten:" Only the Pico-SDLA and FLASH-B WV measuring devices, along with the LOAC and COBALD particle measurement equipment, were flown on March 13, 2012. "

25. *l107: reference for the ETA model?*

    adjusted - There are now references:"At IPMet in Bauru, CAPE values of $\geq 4000\,\mathrm{J\,kg^{-1}}$ were forecast in the central and western parts of São Paulo State by the meso-ETA weather model [Mesinger et al., 2012, Betts and Miller, 1986], of which an adapted version [Held et al., 2007] was routinely running with a horizontal resolution of $10\,\mathrm{km} \times 10\,\mathrm{km}$ during the TRO-Pico campaign."

26. *l116: $1200\,\mathrm{g}$ – > $1.2\,\mathrm{kg}$ as on line 89, for consistency and for the reader to easily recognize which balloon is referred to*

    adjusted - This suggestion has been carried out.

27. *l122-124: relationship between the two measurements?*

– It is unclear to us. During the campaign, one balloon with Pico-SDLA and LOAC was flown, followed by another balloon with FLASH-B and COBALD three hours later. These water vapour profiles measured during TRO-Pico are discussed by Khaykin et al. [2016].

28. *l128: decayed – > decaying?*

adjusted - The sentence has been rewritten:"However, based on a more extensive investigation of a deep convective system that developed during the local afternoon of March 13, 2012, in the southeast of Bauru, and decayed in the evening, the current work provides additional insights into the time evolution of this meteorological state. "

29. *l156: determine – > solve*

adjusted - This suggestion has been carried out:"Furthermore, using a smart grid-nesting system that solves equations simultaneously between computational meshes while applying any number of two-way interactions, the BRAMS/RAMS can solve the fully compressible non-hydrostatic equations [Tripoli and Cotton, 1982]."

30. *l159: reproduces*

adjusted - The sentence has been rewritten:"Marécal et al. [2007] are able to simulate the WV distribution in the tropical UTLS in a deep convective atmosphere using this model."

31. *l161-162: about the simulations of Liu et al (2010): it reads as if these simulations could be very similar to the ones carried out here; more precisions would be welcome. Were these simulations validated against observations? How?*

– Because they use three nested grids with the BRAMS model, these simulations appear to be very comparable to those in Liu et al. [2010]. However, more overshoot case studies using cloud-resolving models are needed to further comprehend their large-scale impact on the stratospheric water budget. Nonetheless, Liu et al. [2010] focus on well-organised convective systems in West Africa, whereas the current work focuses on significantly less well-organised convective systems in South America, with a little greater horizontal resolution (800 m here versus 1 km in Liu et al. [2010]). Furthermore, we have extended on our advantages (S-band radar) in this study to further constrain the simulations.

32. *l168: before the paragraph explaining the technical setup, the modelling strategy (and in particular the overall choices and compromises for the nesting and domains) should be explained*

adjusted - This paragraph has been rewritten:"We use the BRAMS model to run three cloud-resolving simulations, including multiple grid-nesting to explicitly address the stratospheric overshoots associated with the case study in sect. 2. In these simulations, the modelling strategy is to assess the sensitivity of the stratospheric water budget linked to overshoots to the model setup, such as microphysical parameters or vertical resolution, resulting in various hydration or ice injection amounts. It is likely to have an impact on our

conclusions about the underlying physical characteristics related with overshoots, as well as the mechanism for setting them up in large-scale $H_2O$ nudging scheme simulations (or Brazilian size). We employ the same domain (mother-grid) as a step forward from Behera et al. [2018] seasonal scale study, where the model cannot explicitly resolve the overshoots. Then we raise the spatial resolution until we reach the third grid, ensuring that the overshoots are explicitly resolved. We start the simulation several hours before the onset of deep convection activity in the radar data, because we will use Bauru radar observation to evaluate the development of convective cells, as mentioned in sect. 2.3, and to give the model enough time to spin up.

Following that, we run three simulations with a spatial resolution of $800\,m \times 800\,m$. The first of the three simulations is the reference simulation (REF). The shape parameter ($\nu$) of the hydrometeors in the bulk microphysics setting differs from REF in the second simulation, which is indicated as NU21 ($\nu = 2.1$). NU21 is projected to produce hydrometeors with greater mean mass diameters. To better assess TTL dynamics, the third simulation, denoted HVR (High Vertical Resolution) hereafter, has a greater vertical grid-point resolution than REF and NU21. The impact of NU21's sensitivity to the microphysical component, as well as HVR's vertical resolution, on simulations of deep convection and overshooting plumes, is then examined."

33. *l171: presentation?*

    adjusted - We have replaced it with 'resolution'.

34. *l183: this top is rather low given the height of the phenomena of interest; what is the vertical coordinate? What gives confidence to the authors that this model depth is sufficient?*
    adjusted - Marécal et al. [2007], Liu et al. [2010] have given us confidence in our cloud-

    resolving simulation setup. During the TRO-Pico campaign, they employed the BRAMS model with a roughly same setup as we do here to examine overshoots over Brazil. Otherwise, the CSU RAMS model's terrain-following sigma (vertical) coordinate system is well-known. The text has been updated to include this sentence.

    The text is now:"We restrict the top layer of the domain to $30\,km$ altitude with a sponge layer of $5\,km$ to absorb gravity waves at the top on a terrain-following sigma coordinate system, regardless of the vertical resolution of the simulations."

35. *l187: 'which varies between 2 and 10s for the coarsest / exterior grid' ?*

    adjusted - This suggestion has been carried out:"To ensure numerical stability, the simulation integration time step varies between $2\,s$ to $10\,s$ for the coarsest grid."

36. *l204: 'all the three' – > the three?*

    adjusted - This suggestion has been carried out:"We validate the three BRAMS simulations using observations from the S-Band radar of IPMet, located in Bauru, and the balloon-borne measurements of the TRO-Pico campaign, respectively."

37. *l209: interpreting – > comparing?*

adjusted - This suggestion has been carried out:"We examine the BRAMS model's capacity to initiate and describe deep convection activity at an accurate time and location by comparing simulated outputs to S-Band radar data."

38. *l212: 'we determine the cloud top for this range of altitude': ambiguous formulation*

adjusted - The paragraph corresponding to line 212 has been modified:"To do so, we estimate the modelled cloud top layers every 1 km at altitudes ranging from 9 km to 20 km, much like the echo top products. We determine the modelled cloud top height for this altitude range if the concentration of condensed water, i.e., ice plus liquid, exceeds a specified mixing ratio threshold within a specific layer."

39. *l231: is it 'earlier' or 'later'?*

adjusted - This correction has been carried out. 'earlier' is replaced with 'later'.

40. *l237: remove 'now'*

adjusted - This suggestion has been carried out.

41. *l238: with tops typically at 9 to 10 km altitude?*

adjusted - This suggestion has been carried out:"The radar is largely cloud-free at the start of the convective activity (15:01 UT); the only convective cells are around 100 km south-southeast of Bauru near Botucatu, with tops typically at 9 km to 10 km in altitude."

42. *l243-244: By 15:00 UT, the deep convection altitude in HVR is also higher than in REF...*

adjusted - This suggestion has been carried out:"By 15:00 UT, the deep convection altitude in HVR is also higher than in REF and the radar echo tops. "

43. *l249-250: good point, but the formulation is somewhat clumsy; this should be reformulated*

adjusted - This suggestion has been carried out:"Thus, all simulations predict the onset of convective activity to be slightly earlier than observed. Given the uncertainties in modelling and S-band radar perceptions of deep convective activity, associating one-by-one simulations with radar convective cells in spatial and temporal terms is a difficult task [e.g., Li et al., 2008, Rowe and Houze, 2014, Weisman et al., 1997]. As a result, it may not be the most appropriate criterion for evaluating these disorganised deep convective cloud simulations."

44. *l261: higher than REF and NU21: could this be quantified?*

adjusted - To avoid any misunderstanding, the sentence has been rewritten. We're talking about the number of overshoots here. In addition, from the three simulations, we determine the number of overshoots and their cloud top heights in Table 1. According to the estimates, there are 10 overshoots in REF, 6 in NU21, and about 18 in HVR.

The text is now:"HVR, on the other hand, has approximately 18 overshooting plumes during the observation period, which is significantly more than REF (10 overshoots) and NU21 (6 overshoots)."

45. *l267: inertial gravity wave: should this be internal gravity wave?*

adjusted - We believe we are describing the same phenomenon. However, two references for inertial (internal) gravity waves has been added.

The text is now:"Aside from that, Eulerian model simulations of high vertical resolution, high-frequency wave motions, such as inertial-gravity waves [e.g., Staquet, 2004, Young, 2021], can be overdetermined."

46. *l268-269: given that there are only 3 simulations, it is unfortunate to leave one out. Are there not uses that could still be of relevance? (Sensitivity..?)*

adjusted - We are surprised to see such deep convection behaviour in HVR run. We could not help ourselves when we observed the animation of cloud temporal progression compared to the S-band radar and the other two simulations. Despite this, we use HVR to count the number of overshoots (Table 1), a statistic that clearly distinguishes this run from REF and NU21 runs. Apart from that, we do not do any additional analysis with HVR. While there may be more applications for HVR, we have modified this paragraph to include more theoretical information as well as performance references to support our statements.

The updated paragraph is:"To further understand the situation, one can expect HVR to determine more reliable dynamics across the tropical tropopause than REF and NU21, respectively. Contrary to expectations, it tends to intensify massive deep convection activity. A plausible fact to explain such behaviour in HVR is the ratio between vertical and horizontal grid points, which overestimates vertical motions due to grid cell saturation [Homeyer et al., 2014, Homeyer, 2015]. It might be the model's Courant–Friedrichs–Levy(CFL) limit, which in finite-difference simulation techniques constrains the relationship between infinitesimal increases in space grid points and infinitesimal time step increments. In the BRAMS model, the von Neumann stability assessment [Deriaz and Haldenwang, 2020] is necessary for the transport equations related to convection. Aside from that, Eulerian model simulations of high vertical resolution, high-frequency wave motions, such as inertial-gravity waves [e.g., Staquet, 2004, Young, 2021], can be overdetermined. As a result, they can exaggerate cloud microphysics [Aligo et al., 2009] and cause erroneous cloud conditions near the TTL [Jensen and Pfister, 2004]. Therefore, we leave HVR out of the next sections to describe the details, and we do not look at this simulation's water budget in the lower stratosphere."

47. *l366-368 and l372-373: redundant, the second occurrence could be removed*

adjusted - This suggestion has been carried out.

The updated paragraph is:"We provide the five conceivable combinations of hydrometeors inside an overshooting plume to document the quantitative information collected from the simulations on the structural characteristics of a typical overshooting plume. Its base is at the 380 K isentropic level, which is the stratosphere's lowest layer. At the 380 K isentropic

level, the instantaneous mass flux of individual hydrometeors is also estimated. Between 380 K to 430 K isentropic levels, it comprises the estimation of total ice mass and the five types of ice particles. Finally, a table provides the quantities that could lead to a road map of a nudging scheme of the water vapour enhancement in the lower stratosphere due to overshoots in large-scale simulations, which could lead to the quantification of the influence of overshoots on a large scale."

48. *l388-393: very descriptive of model output, but could some more physical interpretation be suggested*

adjusted - This paragraph has been revised to include greater physical meaning and reasoning. However, further physical interpretation of such processes would only be possible and make sense if we could establish a nudging method and run a large-scale simulation using this data.

The text is now:"Within the overshooting plume, Fig. 5 also reveals a large amount of aggregates and graupel at the tropopause level, particularly for REF. It is worth noting that pristine ice is completely absent towards the plume's deepest core at the base (16.6 km height, ∼380 K). Snow, aggregates, and, to a lesser extent, graupels are the only hydrometeors that survive. The major ice hydrometeors in NU21 are snow particles, which disperse across a small area with a radius of around 5 km. The overshooting dome at the edge of the plume near the tropopause level in all three scenarios is entirely formed of pristine ice. In both scenarios going up to 18 km, well into the stratospheric region of the TTL, only pristine ice (70%) and snow (30%) are the principal constituents of the overshooting dome. Graupel and aggregates are present in REF, but not in NU21. This finding is in line with sensitivity tests conducted by manipulating microphysics in Chemel et al. [2009], Wu et al. [2009], who used the WRF model to investigate convective updrafts during the monsoon over Darwin, Australia. Our model illustrates an overshooting plume's overall particle distribution as well as its thermodynamic structure, which is controlled by particle size distribution and affects the convective updraft."

49. *l396: no need for a new paragraph*

adjusted - This suggestion has been carried out.

50. *l399-401: this sentence is not very clear; it could read as a criticism of that previous study, yet that study is by the same authors*

adjusted - This sentence has been rewritten:"It is roughly the grid-point resolution of a large-scale simulation (400 km$^2$), where Behera et al. [2018] show that with such horizontal grid-point resolution, BRAMS cannot explicitly produce overshoots, and illustrate the TTL dynamics and WV variability at a continental scale during a full wet season. In a cloud-resolving scale simulation, BRAMS generates overshoots that spread over 450 km$^2$ in the area at 380 K level, expanding from the third grid to the mother grid to disclose the intensity of convection. Hence, it is a critical point to consider when planning an overshoot nudging scheme."

51. *l501: add 'the simulated': 'the simulated' overshooting plumes reaching...*

adjusted - This suggestion has been carried out.

52. *l513: would it be possible to attempt to translate these numbers into a change of the ppmv content of water vapor, with appropriate assumptions on the volume affected by the injection?*

– A model like this can easily figure out the water vapour mixing ratio, which is largely determined by the background. In addition, the shape and distribution of the overshooting plume will have a substantial impact on the water vapour mixing ratio in the overshoots area. As a result, it cannot be compared to large-scale budget estimates. A mass budget, on the other hand, is an objective output that measures the amount of water injected into the stratosphere or that remains there after the event.

**References**

Wojciech W Grabowski. Coupling cloud processes with the large-scale dynamics using the cloud-resolving convection parameterization (crcp). *Journal of the Atmospheric Sciences*, 58(9): 978–997, 2001.

Marat F Khairoutdinov and David A Randall. A cloud resolving model as a cloud parameterization in the ncar community climate system model: Preliminary results. *Geophysical Research Letters*, 28(18):3617–3620, 2001.

Marat Khairoutdinov, David Randall, and Charlotte DeMott. Simulations of the atmospheric general circulation using a cloud-resolving model as a superparameterization of physical processes. *Journal of the Atmospheric Sciences*, 62(7):2136–2154, 2005.

Marat Khairoutdinov and David Randall. High-resolution simulation of shallow-to-deep convection transition over land. *Journal of Atmospheric Sciences*, 63(12):3421–3436, 2006.

Abhinna K Behera, Emmanuel D Rivière, Virginie Marécal, Jean-François Rysman, Claud Chantal, Geneviève Sèze, Nadir Amarouche, Melanie Ghysels, Sergey M Khaykin, Jean-Pierre Pommereau, et al. Modeling the TTL at Continental Scale for a Wet Season: An Evaluation of the BRAMS Mesoscale Model Using TRO-Pico Campaign, and Measurements From Airborne and Spaceborne Sensors. *Journal of Geophysical Research: Atmospheres*, 123(5):2491–2508, 2018.

J-P Vernier, TD Fairlie, T Deshler, M Venkat Ratnam, H Gadhavi, BS Kumar, M Natarajan, AK Pandit, ST Akhil Raj, A Hemanth Kumar, et al. Batal: The balloon measurement campaigns of the asian tropopause aerosol layer. *Bulletin of the American Meteorological Society*, 99 (5):955–973, 2018.

J-P Pommereau, Anne Garnier, Gerhard Held, AM Gomes, Florence Goutail, Georges Durry, François Borchi, Alain Hauchecorne, Nadège Montoux, P Cocquerez, et al. An overview of the HIBISCUS campaign. *Atmospheric Chemistry and Physics*, 11(5):2309–2339, 2011.

MEE Hassim and TP Lane. A model study on the influence of overshooting convection on TTL water vapour. *Atmospheric Chemistry and Physics*, 10(20):9833–9849, 2010.

Drew Shindell, David Rind, Nambeth Balachandran, Judith Lean, and Patrick Lonergan. Solar cycle variability, ozone, and climate. *Science*, 284(5412):305–308, 1999.

Drew T. Shindell. Climate and ozone response to increased stratospheric water vapor. *Geophysical Research Letters*, 28(8):1551–1554, 2001. ISSN 1944-8007.

RL Herman, K Drdla, JR Spackman, DF Hurst, PJ Popp, CR Webster, PA Romashkin, JW Elkins, EM Weinstock, BW Gandrud, et al. Hydration, dehydration, and the total hydrogen budget of the 1999/2000 winter Arctic stratosphere. *Journal of Geophysical Research: Atmospheres*, 107 (D5):SOL–63, 2002.

Piers M de F Forster and KP Shine. Assessing the climate impact of trends in stratospheric water vapor. *Geophysical research letters*, 29(6):10–1, 2002.

Owen B. Toon, E. V. Browell, S. Kinne, and J. Jordan. An analysis of lidar observations of polar stratospheric clouds. *Geophysical Research Letters*, 17(4):393–396, 1990. ISSN 1944-8007.

ME Hervig, KS Carslaw, Th Peter, T Deshler, LL Gordley, G Redaelli, U Biermann, and JM Russell III. Polar stratospheric clouds due to vapor enhancement: Haloe observations of the antarctic vortex in 1993. *Journal of Geophysical Research: Atmospheres*, 102(D23):28185–28193, 1997.

James R Holton and Andrew Gettelman. Horizontal transport and the dehydration of the stratosphere. *Geophysical Research Letters*, 28(14):2799–2802, 2001.

William J Randel, Fei Wu, Andrew Gettelman, JM Russell III, Joseph M Zawodny, and Samuel J Oltmans. Seasonal variation of water vapor in the lower stratosphere observed in halogen occultation experiment data. *Journal of Geophysical Research: Atmospheres*, 106(D13):14313–14325, 2001.

Steven C Sherwood and Andrew E Dessler. On the control of stratospheric humidity. *Geophysical research letters*, 27(16):2513–2516, 2000.

S Fueglistaler, M Bonazzola, PH Haynes, and Thomas Peter. Stratospheric water vapor predicted from the Lagrangian temperature history of air entering the stratosphere in the tropics. *Journal of Geophysical Research: Atmospheres*, 110(D8), 2005.

R James, M Bonazzola, B Legras, K Surbled, and S Fueglistaler. Water vapor transport and dehydration above convective outflow during Asian monsoon. *Geophysical Research Letters*, 35 (20), 2008.

Samuel J Oltmans, Holger Vömel, David J Hofmann, Karen H Rosenlof, and Dieter Kley. The increase in stratospheric water vapor from balloonborne, frostpoint hygrometer measurements at Washington, DC, and Boulder, Colorado. *Geophysical Research Letters*, 27(21):3453–3456, 2000.

K. H. Rosenlof, S. J. Oltmans, D. Kley, J. M. Russell, E.-W. Chiou, W. P. Chu, D. G. Johnson, K. K. Kelly, H. A. Michelsen, G. E. Nedoluha, E. E. Remsberg, G. C. Toon, and M. P. McCormick. Stratospheric water vapor increases over the past half-century. *Geophysical Research Letters*, 28 (7):1195–1198, 2001.

William J. Randel, Fei Wu, Holger Vömel, Gerald E. Nedoluha, and Piers Forster. Decreases in stratospheric water vapor after 2001: Links to changes in the tropical tropopause and the Brewer-Dobson circulation. *Journal of Geophysical Research: Atmospheres*, 111(D12), 2006.

M. Scherer, H. Vömel, S. Fueglistaler, S. J. Oltmans, and J. Staehelin. Trends and variability of midlatitude stratospheric water vapour deduced from the re-evaluated Boulder balloon series and HALOE. *Atmospheric Chemistry and Physics*, 8(5):1391–1402, 2008.

Dian J. Seidel and William J. Randel. Variability and trends in the global tropopause estimated from radiosonde data. *Journal of Geophysical Research: Atmospheres*, 111(D21), 2006. ISSN 2156-2202.

William J Randel and Eric J Jensen. Physical processes in the tropical tropopause layer and their roles in a changing climate. *Nature Geoscience*, 6(3):169–176, 2013.

J-P Chaboureau, J-P Cammas, J Duron, PJ Mascart, NM Sitnikov, and H-J Voessing. A numerical study of tropical cross-tropopause transport by convective overshoots. *Atmospheric Chemistry and Physics*, 7(7):1731–1740, 2007.

DP Grosvenor, TW Choularton, H Coe, and G Held. A study of the effect of overshooting deep convection on the water content of the TTL and lower stratosphere from Cloud Resolving Model simulations. *Atmospheric Chemistry and Physics*, 7(18):4977–5002, 2007.

Charles Chemel, Maria R Russo, John A Pyle, Ranjeet S Sokhi, and Cornelius Schiller. Quantifying the imprint of a severe Hector thunderstorm during ACTIVE/SCOUT-O3 onto the water content in the upper troposphere/lower stratosphere. *Monthly weather review*, 137(8):2493–2514, 2009.

XM Liu, ED Rivière, V Marécal, G Durry, A Hamdouni, J Arteta, and Sergey Khaykin. Stratospheric water vapour budget and convection overshooting the tropopause: modelling study from SCOUT-AMMA. *Atmospheric Chemistry and Physics*, 10(17):8267–8286, 2010.

Thibaut Dauhut, Jean-Pierre Chaboureau, Juan Escobar, and Patrick Mascart. Large-eddy simulations of Hector the convector making the stratosphere wetter. *Atmospheric Science Letters*, 16(2):135–140, 2015.

Thierry Corti, BP Luo, M De Reus, D Brunner, Francesco Cairo, MJ Mahoney, G Martucci, Renaud Matthey, Valentin Mitev, FH Dos Santos, et al. Unprecedented evidence for deep convection hydrating the tropical stratosphere. *Geophysical Research Letters*, 35(10), 2008.

Sergey Khaykin, J-P Pommereau, L Korshunov, V Yushkov, J Nielsen, N Larsen, T Christensen, Anne Garnier, A Lukyanov, and E Williams. Hydration of the lower stratosphere by ice crystal geysers over land convective systems. *Atmospheric Chemistry and Physics*, 9(6):2275–2287, 2009.

S Iwasaki, T Shibata, H Okamoto, H Ishimoto, and H Kubota. Mixtures of stratospheric and overshooting air measured using A-Train sensors. *Journal of Geophysical Research: Atmospheres*, 117(D12), 2012.

Maryann R. Sargent, Jessica B. Smith, David S. Sayres, and James G. Anderson. The roles of deep convection and extratropical mixing in the tropical tropopause layer: An in situ measurement perspective. *Journal of Geophysical Research: Atmospheres*, 119(21):12,355–12,371, 2014.

Sergey M Khaykin, Jean-Pierre Pommereau, Emmanuel D Riviere, Gerhard Held, Felix Ploeger, Melanie Ghysels, Nadir Amarouche, Jean-Paul Vernier, Frank G Wienhold, and Dmitry Ionov. Evidence of horizontal and vertical transport of water in the Southern Hemisphere tropical tropopause layer (TTL) from high-resolution balloon observations. *Atmospheric chemistry and physics*, 16(18):12273–12286, 2016.

EJ Jensen, Laura L Pan, Shawn Honomichl, Glenn S Diskin, Martina Krämer, Nicole Spelten, Gebhard Günther, Dale F Hurst, Masatomo Fujiwara, Holger Vömel, et al. Assessment of observational evidence for direct convective hydration of the lower stratosphere. *Journal of Geophysical Research: Atmospheres*, 125(15):e2020JD032793, 2020.

Fedor Mesinger, Sin Chan Chou, Jorge L Gomes, Dusan Jovic, Paulo Bastos, Josiane F Busta-mante, Lazar Lazic, André A Lyra, Sandra Morelli, Ivan Ristic, et al. An upgraded version of the eta model. *Meteorology and Atmospheric Physics*, 116(3-4):63–79, 2012.

AK Betts and MJ Miller. A new convective adjustment scheme. Part II: Single column tests using GATE wave, BOMEX, ATEX and arctic air-mass data sets. *Quarterly Journal of the Royal Meteorological Society*, 112(473):693–709, 1986.

Gerhard Held, Jorge Luis Gomes, and EL Nascimento. Forecasting severe weather occurrences in the state of são paulo, brazil, using the meso-eta model. In *Proceedings, 4th European Conference on Severe Storms*, 2007.

GJ Tripoli and WR Cotton. The colorado state university three-dimensional cloud/mesoscale model-1982 part i: General theoretical framework and sensitivity experiments. *J. Rech. Atmos.*, 16:185–219, 1982.

V Marécal, G Durry, K Longo, S Freitas, ED Riviere, and Michel Pirre. Mesoscale modelling of water vapour in the tropical UTLS: two case studies from the HIBISCUS campaign. *Atmospheric Chemistry and Physics*, 7(5):1471–1489, 2007.

Yaping Li, Edward J Zipser, Steven K Krueger, and Mike A Zulauf. Cloud-resolving modeling of deep convection during KWAJEX. Part I: Comparison to TRMM satellite and ground-based radar observations. *Monthly weather review*, 136(7):2699–2712, 2008.

Angela K Rowe and Robert A Houze. Microphysical characteristics of mjo convection over the indian ocean during dynamo. *Journal of Geophysical Research: Atmospheres*, 119(5):2543–2554, 2014.

Morris L Weisman, William C Skamarock, and Joseph B Klemp. The resolution dependence of explicitly modeled convective systems. *Monthly Weather Review*, 125(4):527–548, 1997.

Chantal Staquet. Gravity and inertia-gravity internal waves: Breaking processes and induced mixing. *Surveys in geophysics*, 25(3-4):281–314, 2004.

William R Young. Inertia-gravity waves and geostrophic turbulence. *Journal of Fluid Mechanics*, 920, 2021.

Cameron R Homeyer, Laura L Pan, and Mary C Barth. Transport from convective overshooting of the extratropical tropopause and the role of large-scale lower stratosphere stability. *Journal of Geophysical Research: Atmospheres*, 119(5):2220–2240, 2014.

Cameron R Homeyer. Numerical simulations of extratropical tropopause-penetrating convection: Sensitivities to grid resolution. *Journal of Geophysical Research: Atmospheres*, 120(14):7174–7188, 2015.

Erwan Deriaz and Pierre Haldenwang. Non-linear cfl conditions issued from the von neumann stability analysis for the transport equation. *Journal of Scientific Computing*, 85(1):1–17, 2020.

Eric A Aligo, William A Gallus, and Moti Segal. On the impact of WRF model vertical grid resolution on Midwest summer rainfall forecasts. *Weather and forecasting*, 24(2):575–594, 2009.

Eric Jensen and Leonhard Pfister. Transport and freeze-drying in the tropical tropopause layer. *Journal of Geophysical Research: Atmospheres*, 109(D2), 2004.

Jingbo Wu, Anthony D Del Genio, Mao-Sung Yao, and Audrey B Wolf. Wrf and giss scm simulations of convective updraft properties during twp-ice. *Journal of Geophysical Research: Atmospheres*, 114(D4), 2009.

---

## Author Response (AR1)

Authors' comments on the first revision of the manuscript: "On the cross-tropopause transport of water by tropical convective overshoots: a mesoscale modelling study constrained by in situ observations during TRO-Pico field campaign in Brazil", Reviewer-1

We sincerely thank the reviewer for the pertinent and insightful comments that encouraged us to improve our manuscript. The point-by-point response to the reviewer's comments is documented here. The reviewer's original comments are listed. Italic and boldface fonts are used in the typesets. Each remark is followed by our response. When a change is made to the original version of the manuscript, the word "adjusted" is always included in the response. Unless otherwise specified, the line numbers, page numbers, figure numbers, and table numbers refer to the original version of the manuscript. We have also attached the revised version of the manuscript.

**1 Minor Comments**

1. L. 75: The authors should write the relative humidity with respect to ice.

adjusted - Line 75 of the text has been updated to include the RHi values from Khaykin et al. [2016].

The text is now: "On that particular day, two lightweight balloon-borne hygrometers intercepted a hydrated stratospheric air parcel emanating from two distinct overshooting plumes. However, no ice particles were detected by the particle counter/backscatter sondes. It is also worth noting that at these altitudes, the relative humidity with respect to ice was reported to be about 40-50%."

2. L. 85: Are there several IOPs? If so, the authors should describe whole TRO-pico campaign very briefly.

adjusted - Yes, there are two IOPs in the whole campaign. The paragraph corresponding to line 85 has been changed to eliminate confusion and provide more information about the entire campaign.

The paragraph begins with: "TRO-Pico is a French initiative based on a small balloon campaign in Bauru (22.36°S, 49.03°W), State of São Paulo, Brazil, and funded by the Agence Nationale de la Recherche (ANR). Its purpose is to study the stratospheric water vapour entry in the tropics at different spatial and time scales. In particular, TRO-Pico main's goal is to better quantify the role of overshooting convection at a local scale in order to better quantify its role at a larger scale with respect to other processes. It took place in March 2012 for the first intensive observation period (IOP) and from November 2012 to March 2013, with regular soundings including a second IOP in January and February 2013. The case under investigation in this paper is part of the first IOP while Behera et al. [2018] investigated the November 2012 to March 2013 TRO-Pico period."

**3. L.170: The authors should add the main aim to introduce NU21 for the simulation and/or the characteristics of Eq. 1; it would imply the results of L. 389, L. 435, and L. 487-492.**

adjusted - Please see Fig. 1 in the journal of Atmospheric Research by Walko et al. [1995]. On the other hand, the paragraphs corresponding to lines 155–160 and 170 have been revised to include more information on the shape parameter and the objective of carrying out the NU21 simulation. To better assess TTL dynamics, the third simulation, denoted HVR (High Vertical Resolution) hereafter, has a greater vertical grid-point resolution than REF and NU21. Nonetheless, we have attached a figure (#1#) that illustrates a comparison of REF and NU21 mean mass diameter variations in altitude around the overshoots. As expected, the mean mass diameter in NU21 is slightly greater than that in REF, in particular for pristine ice. We recall here that pristine is the first ice hydrometeor to freeze, so that the comparison between REF and NU21 is more straightforward.

The text is now: "Following that, we run three simulations with a spatial resolution of  $800 \text{ m} \times 800 \text{ m}$ . The first of the three simulations is the reference simulation (REF). The shape parameter ( $\nu$ ) of the hydrometeors in the bulk microphysics setting differs from REF in the second simulation, which is indicated as NU21 ( $\nu = 2.1$ ). NU21 is projected to produce hydrometeors with greater mean mass diameters. To better assess TTL dynamics, the third simulation, denoted HVR (High Vertical Resolution) hereafter, has a greater vertical grid-point resolution than REF and NU21. The impact of NU21's sensitivity to the microphysical component, as well as HVR's vertical resolution, on simulations of deep convection and overshooting plumes, is then examined."

**1# Vertical profiles of mean mass diameters in the vicinity of overshoots for REF and NU21, respectively. The black lines are for REF at 16:15 UT, 22.0°S, 49.18°E. The green lines are for NU21 at 15:45 UT, 22.0°S, 49.2°E (almost the same position but not the same time). Those times correspond to the ones in Fig. 1 of the submitted paper. The positions correspond to the positions of the maximum overshoot in each case.**

**4. L. 226 and Figure 1: The authors should point out "three cells" by arrows in the figure.**

adjusted - Fig. 1 has been updated to include arrows that point to the storm cells.

**2 Specific Comments**

**2.1 Section 6.2 and Figure 8**

1. The authors should define "the total mass budget" clearly; Was it integrated by time and whole area ( $1840 \text{ km} \times 1640 \text{ km}$ )? The authors should add that liquid was neglected.

adjusted - A remark concerning the liquid content has been included to the paragraph corresponding to line 427 of the WV mass budget calculation. It is worth noting that all of the mass budget calculations are limited to the third grid.

The text is now: "Fig. 8 depicts the total mass budget (kilo tonne, kt) for the five types of ice hydrometeors: pristine ice, snow, aggregates, graupel, and hail, as well as water vapour. It is worth mentioning that the amount of liquid in this calculation has no bearing. The simulations' third grid, which has a domain size of  $201 \text{ km} \times 165 \text{ km}$  and isentropic values ranging from 380 K to 430 K, is used for time-integrated estimation. Because none of the convective plumes in the simulations exceed this isentropic level, the maximum level is 430 K."

2. L. 433: The author should write "kilo tons (kt)" because kt is usually used for "knot". The authors should explain how to calculate 8 kt, which is probably "ice+WV at 17:30" – "ice+WV at 15:00."

adjusted - This is exactly how you interpreted it. To avoid any misunderstanding, the mass unit has been explicitly specified in kilotons in line 433, and the text has been revised.

The text is now: "Our mass budget estimation begins with an unperturbed state (zero total mass), i.e., the time before deep convection begins in each simulation, which is 15:00 UT for REF and 14.00 UT for NU21, respectively, and ends at 17:30 UT for both. This is because the WV time evolution reaches a near plateau profile without including any further overshoots, which would otherwise make the study more difficult. Furthermore, the ice profile (dotted red) is descending, indicating that deep convection activity in the model has ended. Simultaneously, the WV profile (dotted blue) rises and settles around 17:30 UT."

3. The authors should explain the legends in Fig. 8. I believe that "17 km .  $\Gamma(v)$  is the normalisation constant, and  $D_n$  is the characteristic diameter of the modified gamma distribution. A bigger v indicates a narrower distribution width and a larger modal diameter. As a result, the proportion of smaller and bigger hydrometeors in the distribution is modulated. The size distribution of hydrometers would be more peaked as the modal diameter increased."

Nonetheless, we have included a figure (#1#) that compares REF and NU21 mean mass diameter variability in altitude around overshoots for pristine ice, snow, and aggregates. As expected, NU21 has a slightly larger mean mass diameter than REF, at least for pristine ice, which is the first ice hydrometeor to be formed by freezing. Other hydrometeors have a more complicated conclusion because they arise from pre-existing ice particles. We should also emphasise that this comparison is essentially indicative; because the cells are not identical, they are not strictly comparable one to one.